Manuscript prepared for Geosci. Model Dev.
with version 2015/04/24 7.83 Copernicus papers of the LaTeX class copernicus.cls.
Date: 7 June 2016

# Numerical framework and performance of the new multiple phase cloud microphysics scheme in RegCM4.5: precipitation, cloud microphysics and cloud radiative effects.

Rita Nogherotto[1], Adrian Mark Tompkins[1], Graziano Giuliani[1], Erika Coppola[1], and Filippo Giorgi[1]

[1]The Abdus Salam International Centre for Theoretical Physics ICTP, Trieste
[1]Strada Costiera 11, 34151 Trieste Italy

*Correspondence to:* Rita Nogherotto (rnoghero@ictp.it)

**Abstract.** We implement and evaluate a new parameterization scheme for stratiform cloud microphysics and precipitation within the regional climate model RegCM4. This new parameterization is based on a multiple phase one-moment cloud microphysics scheme built upon the implicit numerical framework recently developed and implemented into the ECMWF operational forecasting model. The parameterization solves five prognostic equations for water vapour, cloud liquid water, rain, cloud ice and snow mixing ratios. Compared to the pre-existing scheme, it allows a proper treatment of mixed-phase clouds and a more physically realistic representation of cloud miscrophysics and precipitation. Various fields from a 10-yr-long integration of RegCM4 run in tropical band mode with the new scheme are compared with their counterparts using the previous cloud scheme and are evaluated against satellite observations. In addition, an assessment using the Cloud Feedback Model Intercomparison Project (CFMIP) Observational Simulator Package (COSP) for a 1-yr sub-period provides additional information for evaluating the cloud optical properties against satellite data. The new microphysics parameterization yields an improved simulation of cloud fields and in particular it removes the overestimation of upper level clouds characteristic of the previous scheme, increasing the agreement with observations and leading to an amelioration of a long-standing problem in the RegCM system. The vertical cloud profile produced by the new scheme leads to a considerably improvement of the representation of the longwave and shortwave components of the cloud radiative forcing.

## 1 Introduction

Despite the recent increase in computing power, the wide range of temporal and spatial scales involving cloud processes still requires parameterizations to allow the representation of clouds in current Global and Regional Climate Models (GCMs and RCMs, respectively). Convective clouds are rep-

resented by cumulus parameterizations, which mostly focus on dynamical and thermodynamical processes and treat the cloud microphysics in simplified ways. Stratiform, or resolved scale, clouds are represented by parameterizations employing more detailed treatments of cloud microphysics through the explicit prognostic simulation of one or more hydrometeors.

Simpler microphysics schemes treat the cloud water prognostically and precipitating water diagnostically (e.g. Rotstayn, 1997; Pal et al., 2000). Observational data show that between -23 °C and 0 °C the occurrence of supercooled water is not negligible (Matveev, 1984), and liquid and ice particles can co-exist for hours and sometimes even days (e.g. Korolev et al., 2003; de Boer et al., 2009). Often cloud schemes diagnose the fraction of cloud water in the ice phase based on the local temperature (e.g. DelGenio et al., 1996). The diagnostic partitioning of cloud water into the liquid and ice phases assumes implicitly that processes within the cloud are fast compared to the model time step, i.e. that the cloud variables are always in equilibrium. Therefore, a diagnostic representation is unable to describe the temporal variability and evolution of mixed-phase clouds and a prognostic treatment of cloud ice and water is necessary to represent the microphysical processes of the two phases (including their contrasting sedimentation rates). More complex microphysics schemes have been therefore introduced to treat separately the cold and warm cloud microphysics by solving prognostic equations for cloud liquid water and ice (e.g. Fowler et al., 1996; Lohmann and Roeckner, 1996). These schemes are especially important as climate models approach resolutions at which cloud physics processes, including convection, need to be explicitly described without the use of parameterization schemes (e.g. Prein et al. 2015). Recently, several studies have illustrated the importance of using a more realistic representation of cloud microphysics in climate models. For example, Cesana et al. (2015) and Komurcu et al. (2014) showed that climate models tend to underestimate the supercooled liquid clouds and models that prognose separately the liquid and ice mixing ratio give a better representation of cloud properties. The Regional Climate Model RegCM version 4 (or RegCM4) of the International Centre for Theoretical Physics (ICTP) is a widely used system that has been applied to local and regional seasonal forecasting and climate change problems for all regions of the globe (e.g. Sylla et al., 2010; Diro et al., 2012a, b; Nogherotto et al., 2013; Coppola et al., 2014; Fuentes-Franco et al., 2014). The model has a wide choice of physical parameterizations for processes such as deep convection, but, to date, uses a simple diagnostic stratiform cloud scheme with a single prognostic cloud variable (Pal et al., 2000). There is a need not only to improve the representation of the cloud processes in the RegCM modelling system, but also to conduct a comprehensive evaluation of the simulated clouds in RegCM integrations, which have received limited attention relative to the surface climate of the model.

In this paper we first present a description of the revised numerics and microphysics of the new 5-phase prognostic parameterization scheme for stratiform clouds. The scheme is then tested in a series of experiments with the RegCM4 run using the tropical band configuration of Coppola et al. (2012), which allows an analysis of the scheme's performance in different climatic settings. The

cloud variables are compared to the existing RegCM4 Subgrid Explicit Moisture Scheme (SUBEX) scheme, and the new parameterization is also assessed using the recently available Cloud Feedback Model Intercomparison Project (CFMIP) Observational Simulator Package (COSP, version 1.3.2) (Bodas-Salcedo et al., 2011), which allows for direct comparison with a range of cloud-relevant satellite products, using model variables in a forward radiative transfer calculation to avoid uncertainties in retrieval techniques. The final section summarizes the findings and makes suggestions for future developments of the scheme.

## 2  Methodology

### 2.1  Regional climate model

The new cloud microphysics parameterization is introduced into the International Centre for Theoretical Physics (ICTP) Regional Climate Model RegCM version 4. RegCM4 is a three-dimensional compressible, hydrostatic, primitive equation atmospheric model based on the dynamics of the NCAR mesoscale model Version 5 (MM5; Grell et al. 1994) and described in Giorgi et al. (2012). In the current version of RegCM4 the resolved scale cloud microphysics is treated by the Subgrid Explicit Moisture Scheme (SUBEX, Pal et al. 2000), which calculates fractional cloud cover as a function of grid point average relative humidity and includes only one prognostic equation for cloud water. Rain is calculated diagnostically and it forms when the in-cloud liquid water exceeds a temperature-dependent threshold (autoconversion). Rain is assumed to fall instantaneously within the model's time step and to grow by accretion of cloud droplets. SUBEX does not treat cold cloud microphysics and the fraction of ice is diagnosed as a function of temperature in the RegCM4 radiation scheme from radiative transfer calculations (Giorgi et al., 2012). The diagnostic split of ice and liquid water assumes that below -30 °C clouds consist of ice and for temperatures above -10 °C clouds are liquid only. This representation is an augmentation of an earlier scheme (Giorgi et al. 1993 which was in turn a simplified version of the scheme of Hsie and Anthes 1984).

### 2.2  New microphysics cloud scheme

The new cloud microphysics scheme considers cloud ice as a separate prognostic variable and also solves prognostic equations for rain and snow, accounting for the major microphysical pathways between these categories (Fig. 1). The scheme includes four hydrometeors in total: cloud liquid water and ice, rain and snow. Each variable is expressed in terms of the grid-mean mixing ratio $q_x$ (kg kg$^{-1}$) and the governing equations for the mass mixing ratios of water vapour $q_v$, cloud water $q_c$, cloud ice $q_i$, rain $q_r$ and snow $q_s$ take the form:

$$\frac{Dq_x}{Dt} = S_i + \frac{1}{\rho}\frac{\partial}{\partial z}(\rho V_x q_x) + D, \tag{1}$$

where $S_i$ includes the microphysical source and sink terms for each hydrometeor, representing the conversion of water substance between microphysical categories (see Figure 1). The second term on the right hand side represents the source of the variable $q_x$ from the layer above due to gravitational sedimentation of the species falling with terminal velocity $V_x$. The substantive derivative on the LHS indicates that the prognostic equations include advection by the large-scale wind. Horizontal and vertical advection become increasingly important as horizontal resolution increasing over time. The $D$ term on the RHS represents any transport or source/sink terms due to the other diabatic processes that are parameterized in the model, such as diffusion or deep convection. The five

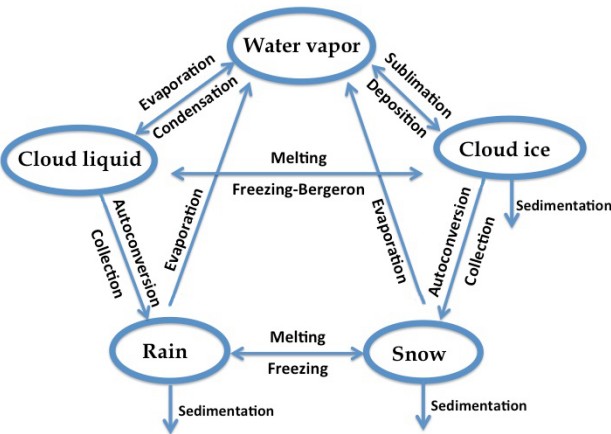

**Figure 1.** Sketch of the new scheme, showing the five prognostic variables and how they are related to each other through microphysical processes.

prognostic equations for the individual species are solved using a simple forward-in-time, implicit solver approach that was implemented in the European Centre for Medium Range Weather Forecasts (ECMWF) integrated forecasting system (IFS) in cycle 31R1 (September 2006) with the objective of reducing the vertical-resolution sensitivity of the earlier explicit solver Tompkins (2005b). Tompkins subsequently generalized the IFS scheme to five species similar to the scheme presented here, which became operational in ECMWF cycle 36r4 (November 2010, Forbes et al., 2011). The scheme has the advantage of being conservative, numerically economical, stable at all timesteps, and employs a numerical solution framework that is trivially expandable to a larger numbers of microphysical variables, facilitating the future representation of hail and graupel categories, or various ice crystal size bins. However, Tompkins (2005b) highlights that the scheme is highly diffusive for fast falling species. Following Tompkins (2005b), the equations are solved using the upstream approach, which utilises the forward difference quotient in time and the backward difference quotient in space. For the time step $n$, dropping the large-scale advection and diabatic contributions as these terms are handled

elsewhere in the model outside the microphysics scheme, the discretized equations are:

$$\frac{q_x^{n+1} - q_x^n}{\Delta t} = A_x + \sum_{y=1}^{m} B_{xy} q_y^{n+1} - \sum_{y=1}^{m} B_{yx} q_x^{n+1} + \frac{\rho_{k-1} V_x q_{x,k-1}^{n+1} - \rho V_x q_x^{n+1}}{\rho \Delta z}. \tag{2}$$

It is seen that the microphysical pathways have been divided between two terms $A$ and $B$, according
to the timescale of the process they describe. Processes that are considered to be fast relative to
the model time step, where the rate term can change substantially over the course of a timestep
(e.g. autoconversion), are treated implicitly and are included in the matrix $B$. A positive term $B_{xy}$
represents a process which is a source of $q_x$ and a sink of $q_y$. Thus $B$ is positive-definite off the

diagonal, with $B_{xx} = 0$ by definition. On the other hand, processes that evolve slowly and can or
should be assumed constant over a model time step (e.g. condensation by large-scale ascent) are
treated explicitly and are included in the matrix $A$ whose elements $A_x$ represent the net contribution
to the variable $q_x$ by the explicit processes. We note that there is not a definitive justification for
how microphysical processes are allocated to each solution category. As sedimentation is in the

downwards direction and there is no transport within the cloud scheme in the upward direction, the
equations can be simply integrated one layer at a time from the top to the bottom layer of the model,
making the solution numerically efficient as in each layer the solution of a $m \times m$ matrix equation
is required, where $m$ is the species number. An $m = 3$ category system at model level $k$ is given by:

$$\begin{pmatrix} 1 + \Delta t(\frac{V_1}{\Delta z} + B_{21} + B_{31}) & -\Delta t B_{12} & -\Delta t B_{13} \\ -\Delta t B_{21} & 1 + \Delta t(\frac{V_2}{\Delta z} + B_{12} + B_{32}) & -\Delta t B_{23} \\ -\Delta t B_{31} & -\Delta t B_{32} & 1 + \Delta t(\frac{V_3}{\Delta z} + B_{13} + B_{23}) \end{pmatrix} \cdot \begin{pmatrix} q_1^{n+1} \\ q_2^{n+1} \\ q_3^{n+1} \end{pmatrix} =$$

$$= \begin{pmatrix} q_1^n + \Delta t\left(A_1 + \frac{\rho_{k-1} V_1 q_{1,k-1}^{n+1}}{\rho \Delta z}\right) & q_1^n + \Delta t\left(A_2 + \frac{\rho_{k-1} V_2 q_{2,k-1}^{n+1}}{\rho \Delta z}\right) & q_3^n + \Delta t\left(A_3 + \frac{\rho_{k-1} V_3 q_{3,k-1}^{n+1}}{\rho \Delta z}\right) \end{pmatrix}$$

where the index $k-1$ represents the layer lying above the solution layer. Unlike implicit terms,
explicit terms can possibly reduce a cloud variable to zero or negative values. In order to avoid this,
and therefore to ensure that all variables remain positive definite at the end of the time step, the initial
vector $\boldsymbol{A}$ containing the explicit source and sink terms is generalised using an anti symmetric matrix
$A$, whose elements $A_{xy} > 0$ represent a source for the variable $q_x$ and a sink for $q_y$:

$$\begin{pmatrix} A_{11} & A_{21} & A_{31} \\ -A_{12} & A_{22} & A_{32} \\ -A_{13} & -A_{23} & A_{33} \end{pmatrix}$$

All the terms in the diagonal, $A_{xx}$, represent microphysical source that are considered "external" to

the scheme, such as the cloud water detrainment from the (mass-flux) shallow and deep convection
schemes. For each time step, before calling the solvers, the sum of all sinks of each variable is
scaled to avoid negative values, a method that avoids negative values while guaranteeing total water
conservation. For each microphysical pathway the change of phase is associated with a release or
absorption of latent heat. Regarding the enthalpy budget, rather than summing the microphysics

pathways (as in the schemes of Tiedtke, 1993; Swann, 1994, for example), which can easily give rise
in coding errors and resulting non-conservation when modifying microphysical parameterizations in
operational and/or evolving models, the source/sink is calculated using the explicit conservation of
the liquid water temperature $T_L$ defined as:

$$T_L = T - \frac{L_v}{C_p}(q_l + q_r) - \frac{L_s}{C_p}(q_i + q_s). \tag{3}$$

Since $\frac{dT_L}{dt} = 0$, the rate of change of the temperature is given by the equation:

$$\frac{\partial T}{\partial t} = \sum_{x=1}^{m} \frac{L_x}{C_p}\left(\frac{dq_x}{dt} - D_{q_x} - \frac{1}{\rho}\frac{\partial}{\partial z}(\rho V_x q_x)\right) \tag{4}$$

where $L_x$ is the latent heat (of fusion or evaporation depending on the processes considered), $D_{q_x}$ is
the convective detrainment and the third term in the brackets is the sedimentation term. We subtract
the convective detrainment term $D_{q_x}$ and the advective flux terms to the rate of change of species

$q_x$ (due to all the processes) because they represent a net $T_L$ flux not associated with latent heating
with changes of phase of water in the scheme itself.

### 2.2.1   Microphysics

a) *Cloud cover*
Unlike the ECMWF IFS, the RegCM4 cloud fraction is not prognostic, but rather uses a diagnostic

approach which has the advantage of simplifying the implementation and numerical cost, but has
a number of disadvantages. The fractional cloud cover $C$ is calculated following the semiempirical
cloudiness parameterization developed by Xu and Randall (1996), which uses the large-scale relative
humidity $RH$ and average condensate (cloud water and cloud ice) mixing ratios $\bar{q}_l = q_l + q_i$ to give
implicit information concerning the subgrid-scale total water distribution (see review in Tompkins,

2002) and the resulting cloud cover:

$$C = \begin{cases} RH^p[1 - \exp(-\frac{\alpha_0 \bar{q}_l}{[(1-RH)q_s]}^\gamma] & \text{if } RH < 1 \\ 1 & \text{if } RH \geq 1 \end{cases} \tag{5}$$

where $p$, $\alpha_0$ and $\gamma$ are determined empirically and their values are 0.25, 0.49 and 100 respectively.
In theory, such a scheme also incorporates the impact of sub-grid temperature variability on cloud
fraction, since temperature fluctuations are implicitly incorporated into the statistics of the cloud re-

solving model simulations to which the scheme is fitted, however temperature fluctuations are likely
underestimated in the small 2D domains used in Xu and Randall (1996), although Tompkins (2005a)
showed that temperature variability is in general far less important relative to total water variability
above the boundary layer. One key disadvantage of using a diagnostic cloud fraction approach is that
the treatment of ice supersaturation in the clear part of the model grid box at temperatures below -

38C, such as in the scheme of Tompkins (2007) is not permitted. This is because standard $RH$ based
schemes (Sundqvist et al., 1989; Xu and Randall, 1996, e.g.) diagnose overcast conditions when the

gridbox is saturated. Modifying the diagnostic relation to introduce a higher threshold for nucleation at cold temperatures (Koop et al., 2000) would not be able to represent the hysteresis between pre and post ice nucleation, in other words, a separate memory is required of where in the grid box nucleation has occurred. The Tompkins (2007) scheme was able to use the prognostic cloud fraction to accomplish this by assuming the nucleation and subsequent ice crystal diffusive growth timescales was fast compared compared to the model timestep, thus assuming precisely ice saturated conditions in the cloudy portion of the grid box. As stated by Tompkins (2007), this is very good assumption if ice nucleation is predominately homogeneous in nature, although even if heterogeneous nucleation predominates it is still reasonable, since to cut off homonucleation completion IN concentrations need to be of an order of magnitude that results in the growth timescale is similar to a typical global model timestep (Kärcher and Lohmann, 2002).

b) *Condensation and evaporation*

The formation of stratiform clouds associated with large-scale lifting of moist air or with radiative cooling is treated as a function of the variation in time of the saturation mixing ration, following Tiedtke (1993). In fact if the saturation mixing ratio decreases, condensation occurs while as it increases evaporation takes place. The variation in time of the saturation mixing ratio can be written as:

$$\frac{dq_{sat}}{dt} = \frac{\partial q_{sat}}{\partial T}\frac{\partial T}{\partial t}\Big|_{diab} + \frac{\partial q_{sat}}{\partial p}\Big|_{ma}\omega \tag{6}$$

This equation shows that the rate of change of the saturation mixing ratio is linked to diabatic cooling $(\partial T/\partial t)_{diab}$ and to the vertical motion with a grid mean vertical velocity $\omega$, where $(\partial q_{sat}/\partial p)_{ma}$ is the variation of $q_{sat}$ along a moist adiabat.

Condensation occurs when:

$$\frac{dq_{sat}}{dt} < 0 \tag{7}$$

The condensation rate $C_1$ is proportional to the amount of cloud and is equal to:

$$C_1 = -C\frac{dq_{sat}}{dt}, \quad \frac{dq_{sat}}{dt} < 0 \tag{8}$$

and all the increase of cloud amount is a source of cloud water unless the process occurs within cold clouds, in which case deposition occurs and ice forms. Due to the diagnostic treatment of the cloud fraction, homogeneous freezing takes place and removes any supersaturation instantaneously.

The scheme treats two processes that induce evaporation: the large scale descent and the diabatic heating, giving rise to $E_1$, and the turbulent mixing of cloud air with drier environmental air, producing $E_2$, so that the total evaporation $E$ is given by:

$$E = E_1 + E_2 \tag{9}$$

As opposed to condensation, evaporation is proportional to the increase of the saturation mixing ratio and to the amount of cloud following:

$$E_1 = C\frac{dq_{sat}}{dt}, \quad \frac{dq_{sat}}{dt} > 0 \tag{10}$$

It is reasonable to assume that the cloud water content within clouds is homogeneously distributed in the horizontal direction, therefore the evaporation is not changing the cloud cover until it reduces to zero.

A very simple treatment of turbulence mixing is adopted in this first version of the scheme that duplicates the approach of Tiedtke (1991) by treating turbulence as a sink of cloud water. As discussed by Tompkins (2002) and Tompkins (2005a), the sign of the turbulent impact on cloud water is only correct if the total water mixing ration $q_t = q_l + q_i$ is smaller than the saturation mixing ratio, otherwise mixing leads to an increase in cloud water. The intention is to correct this when a PDF-based cloud cover parametrization is later implemented.

c) *Condensation from detrainment*

As an input from the convection scheme the microphysics scheme receives the detrained mass flux $D$ that is assumed to condense into cloud water or into ice diagnostically using a coefficient $\alpha$, function of temperature. This process is applied for all types of convection, namely deep, shallow and mid-level and represents an important extension of the model's cumulus parameterization.

The source of water/ice cloud content is given by:

$$\frac{\partial q_x}{\partial t} = \alpha(T)D_x \tag{11}$$

where $x$ represents either ice or liquid according to the value of a function of the temperature $\alpha(T)$.

d) *Autoconversion*

Autoconversion is the mechanism by which rain or snow droplets form from the aggregation of cloud water or ice particles. This process plays a crucial role in the development of precipitation. For this reason we have implemented four different parameterizations of the process, all following the form:

$$P = P_0 \cdot T \tag{12}$$

where $P$ is the autoconversion rate, $P_0$ the autoconversion rate once the autoconversion has started, and $T \leq 1$ is a function that describes the threshold behaviour of this process (Liu and Daum, 2004). The four parameterizations of autoconversion in the scheme, which can be selected by the user, employ different threshold functions: an "all-or-nothing" approach, described in Kessler (1969)

$$\frac{\partial q_r}{\partial t} = k \cdot (q_l - q_{crit}), \quad (\text{with} \quad k = 10^{-3}s^{-1} \quad \text{and} \quad q_{crit} = 0.5 \quad gm^{-3}) \tag{13}$$

and three exponential approaches using smooth threshold functions.

The first following Sundqvist (1978):

$$\frac{\partial q_r}{\partial t} = c_0 F_1 q_l \left\{ 1 - \exp\left[ -\left(\frac{q_l^{cld}}{q_l^{crit}}\right)^2 \right] \right\}, \quad (\text{with} \quad F_1 = 1 + b_1\sqrt{P_{loc}}) \tag{14}$$

where $c_0 = 1.67 \cdot 10^{-4}$ s$^{-1}$, $b_1 = 100$ (kg m$^{-2}$s$^{-1}$) and $P_{loc}$ is the local cloudy precipitation rate.

The second parameterization follows Beheng (1994):

$$\frac{\partial q_r}{\partial t} = c_b \cdot q_l^{3.3}, \quad \text{(where} \quad c_b = 2.461 \cdot 10^5 s^{-1}) \tag{15}$$

and the third following Khairoutdinov and Kogan (2000):

$$\frac{\partial q_r}{\partial t} = c_{kk} \cdot q_l^{2.47}, \quad \text{(where} \quad c_{kk} = 0.355 s^{-1}). \tag{16}$$

The autoconversion of cloud droplets distinguishes between maritime and continental clouds by
240 considering two different values for the cloud droplet concentration number $N$ (Beheng, 1994). The parameterization used for autoconversion of ice follows Equation 14 but with different parameters more appropriate for ice particles with a coefficient $c_0$ that is a function of the temperature T (Lin et al., 1983):

$$c_0 = 10^{-3} \exp(0.025 \cdot (T - 273.15)) \tag{17}$$

Here, the default autoconversion parameterization is set to the Sundqvist?s scheme (eq. 14) and sensitivity studies using different autoconversion schemes need to be carried out for specific applications.

e) *Freezing and melting*

The parameterization of ice crystal nucleation is very simple and takes into account only the ho-
250 mogeneous process, with the ice number concentration ($N_i$) diagnosed according to Meyers et al. (1992). For temperature below the homogeneous nucleation threshold of -38°C, water droplets are assumed to freeze instantaneously. For temperature above this threshold supercooled water and ice are allowed to coexist, they are assumed to be well mixed and are distributed uniformly through the cloud. At temperatures below this threshold the liquid water is assumed to freeze instantaneously
and the process is a source of cloud ice. The ice crystal is then assumed to grow at the expense of the water droplets through the Wegener-Bergeron-Findeisen process following Rotstayn et al. (2000). The melting of ice and snow is parameterized taking into account also the cooling due to the evaporation of liquid water during the melting process. Therefore, the wet-bulb temperature is used instead of the dry-bulb one. Melting occurs if the wet-bulb temperature is greater than 0°C. The
part of the box containing precipitation is allowed to cool to $T_{melt}$=0 °C over a time scale $\tau$. The wet-bulb temperature $T_w$ is parameterized through a numerical approximation suggested by Wilson and Ballard (1999). All rain freezes in a time step if the temperature is lower than 0°C. This process represents a sink for rain and a source for snow. Since freezing would lead to an increase of temperature due to the latent heat release the scheme ensures that the temperature does no exceed
the 0°C threshold. For a more detailed description of the parameterization of microphysical processes we refer the reader to the IFS Documentation, Cy40r1, Part IV: Physical Processes (online at https://software.ecmwf.int/wiki/display/IFS/Official+IFS+Documentation).

### 2.2.2 Simulation experiments

Table 1 describes the simulation experiments conducted and analysed in this work. We completed two 10-year simulations: one using the SUBEX scheme of the standard RegCM4 (hereafter referred to as "SUB") and one with the newly implemented microphysics cloud scheme (hereafter referred to as "MIC"). Both simulations begin on 1 January 2000 and end on 1 January 2010. However the first 5 months of the simulation, i.e. up to May 2000, are not included in the analysis as initial

spin-up period. As already mentioned, in order to obtain a general overview of the model's ability in representing clouds for different climate settings, the model is run over a tropical band domain, (180°W-180°E, 47°S-47°N), as in Coppola et al. (2012) with an horizontal resolution of 90 km and 23 vertical sigma levels. Initial and lateral (north and south) boundary conditions are obtained from the ERA-Interim 0.75°x0.75° reanalysis (Simmons et al., 2007; Dee et al., 2011). Among the many

physics schemes available in RegCM4 (Giorgi et al., 2012), in this study we use the mass flux convection scheme of Tiedtke (1989), the Biosphere-Atmosphere Transfer Scheme, BATS (Dickinson et al., 1993) for land surface processes and the boundary layer scheme of Bretherton et al. (2004), which provides a realistic representation of stratocumulus-capped boundary layers. For a more detailed understanding of the impact of the new scheme on the representation of clouds and cloud

radiative forcing (CRF) the results of a 1-year test run (2007) are analysed and compared to observations using the Cloud Feedback Model Intercomparison Project Observational Simulator Package COSP (Bodas-Salcedo et al., 2011) for both the SUB and MIC model configurations. The monthly mean COSP fields are produced from each 6-hourly RegCM4 output and then averaged over the months and seasons. This analysis is limited to one year, as in most previous studies (e.g. Franklin et al. 2013 and Sud et al. 2013), because of the large amount of processing it requires.

**Table 1.** Description of simulation experiments.

| Simulation Experiments | Descriptions | Year Analyzed |
|---|---|---|
| SUBEX run (SUB) | RegCM4 with baseline cloud physics | 10 |
| MICROPHYSICS run (MIC) | RegCM4 with the new cloud microphysics scheme | 10 |
| SUB run with COSP simulator | Cloud properties from ISCCP, CALIPSO, MISR simulator | 1 |
| MIC run with COSP simulator | Cloud properties from ISCCP, CALIPSO, MISR simulator | 1 |

## 3 Results

In this section we compare the SUB and MIC simulations for precipitation, total cloud cover, vertical cloud distribution and CRF. The model output is assessed against different observational datasets, with focus on the two extreme seasons, December-January-February (DJF) and June-July-August

(JJA).

### 3.1 Precipitation

An unambiguous assessment of the effect of the new scheme on precipitation performance is extremely difficult. On the one hand, the simulation of precipitation is sensitive to the use of different physics schemes in the model, with this sensitivity depending on region and season (e.g. Giorgi et al.
2012; Coppola et al. 2014). On the other hand, observed precipitation in tropical regions is characterized by a substantial level of uncertainty (e.g. Nikulin et al. 2012, Sylla et al. 2013). It is thus likely that the MIC and SUB schemes might show different performances when used with different sets of model configurations or compared with different observation datasets. Exploring this sensitivity would require a large multi-physics model ensemble which is beyond the purpose of the present
paper. Rather, the more limited objective of this section is to illustrate the effect of the MIC scheme with respect to the SUB one within the framework of a model configuration yielding a realistic precipitation simulation in tropical-band mode. Figure 2 shows the DJF and JJA 10-yr precipita-

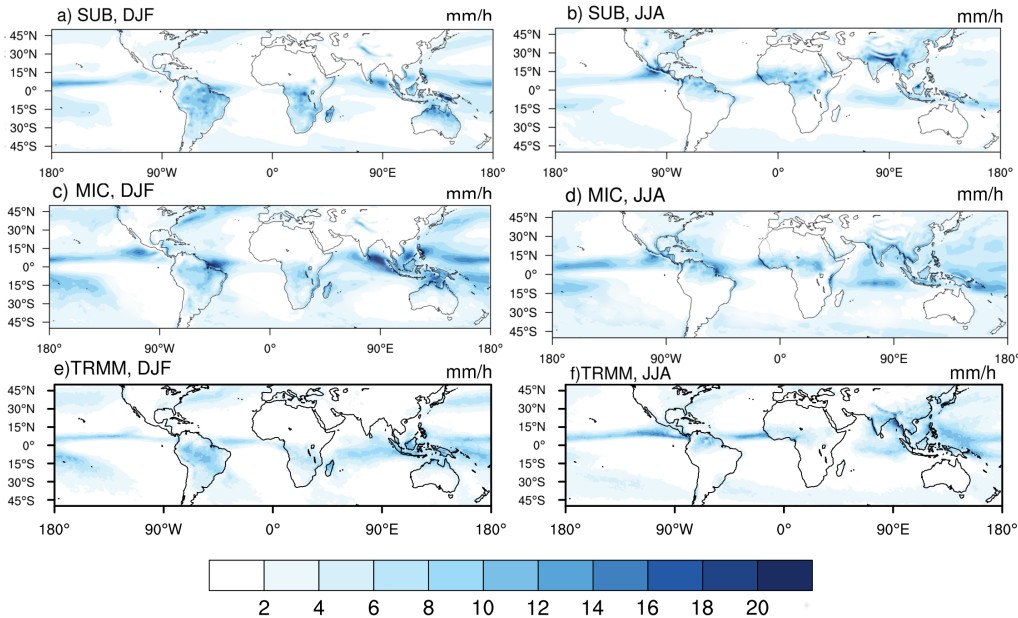

**Figure 2.** Simulated 10-yr mean precipitation (mm h$^{-1}$) for DJF (left), JJA(right) in SUB and MIC runs (top 2 panels) and OBS (bottom); TRMM data represent OBS.

tion climatologies for the SUB and MIC runs along with the corresponding precipitation patterns in the Tropical Rainfall Measuring Mission (TRMM) (Huffman et al., 2007) observation dataset.
Both model configurations produce a good spatial representation of the Intertropical Convergence Zone (ITCZ) and South Pacific Convergence Zone (SPCZ) with maxima in convective precipitation generally following observations. Also captured are the mid-latitude winter storm tracks over the

Atlantic and Pacific mid-latitudes, as well as the main monsoon regions of South America, Africa, India and East/South-east Asia. Overall, the main difference between the two schemes is that the MIC tends to be wetter than the SUB over the oceans and drier over the continental masses. For the present model configuration and in comparison with the TRMM data, this tends to yield an improved agreement with observations over land and a deterioration over oceans. As already mentioned, this conclusion likely depends on the model configuration, however it is clear from Figure 2 that the new microphysics produces a realistic simulation of precipitation, particularly over land, throughout the tropics and sub-tropics. It should also be mentioned that the MIC scheme itself is sensitive to different parameters affecting the production of precipitation, and in particular the ice and snow fall speed and the choice of the autoconversion threshold (Nogherotto, 2015).

## 3.2 Cloud fractions

In this section we present an analysis of the cloud fractional cover. This is accomplished by applying the COSP postprocessing tool to the model output to produce cloud variables comparable to those observed. As already mentioned, this post-processing was carried out only for the seasons December 2006 to February 2007 (DJF) and June to August 2007 (JJA), following the evaluation of clouds in the ACCESS model by Franklin et al. (2013). Total cloud fractions are calculated by the model using the approach of Xu and Randall (1996) and the max-random overlap assumption. The evaluation of total cloud cover is carried out using the GCM simulator-oriented International Satellite Cloud Climatology Project ISCCP cloud product (Pincus et al., 2012), which was prepared to facilitate the evaluation of the model simulated clouds within the framework of the Cloud Feedback Model Intercomparison Project (http://climserv.ipsl.polytechnique.fr/cfmip-obs). Data are averaged over the JJA and DJF 2007 seasons during the daytime, at a horizontal resolution 2.5°x2.5°. Figure 3 shows the total cloud cover in the SUB and MIC simulations for the selected seasons, postprocessed with COSP's ISCCP simulator. These are compared with the corresponding observed ISCCP total cloud amounts for the same seasons (Fig. 3e and Fig. 3f). The ISCCP's observed total cloud fraction averaged over the domain is 66.07% and 64.66% for DJF and JJA, respectively. These values are 68.44% in DJF and 65.35% in JJA for the SUB run, and 61.52% in DJF and 60.04 % in JJA for the MIC (Table 2). Therefore, the SUB scheme produces generally larger cloud fractions than the MIC, and the observations lie within the two model configuration data. In general, both schemes capture the horizontal distribution of clouds over the band domain in both seasons, with maximum cloud cover over the ITCZ and the mid-latitude storm track regions of both hemispheres. However an analysis of the spatial correlation between the two schemes and the observations reveals that the new parameterization improves the horizontal distribution of clouds (Table 2): while the SUB scheme tends to overestimate the magnitude and extension of total cloud amounts across the ITCZ, the MIC scheme shows a slight underestimation but it improves the stratiform cloud cover between 30 and 45 °S, yielding higher spatial correlation values compared to those obtained with SUB. For a more

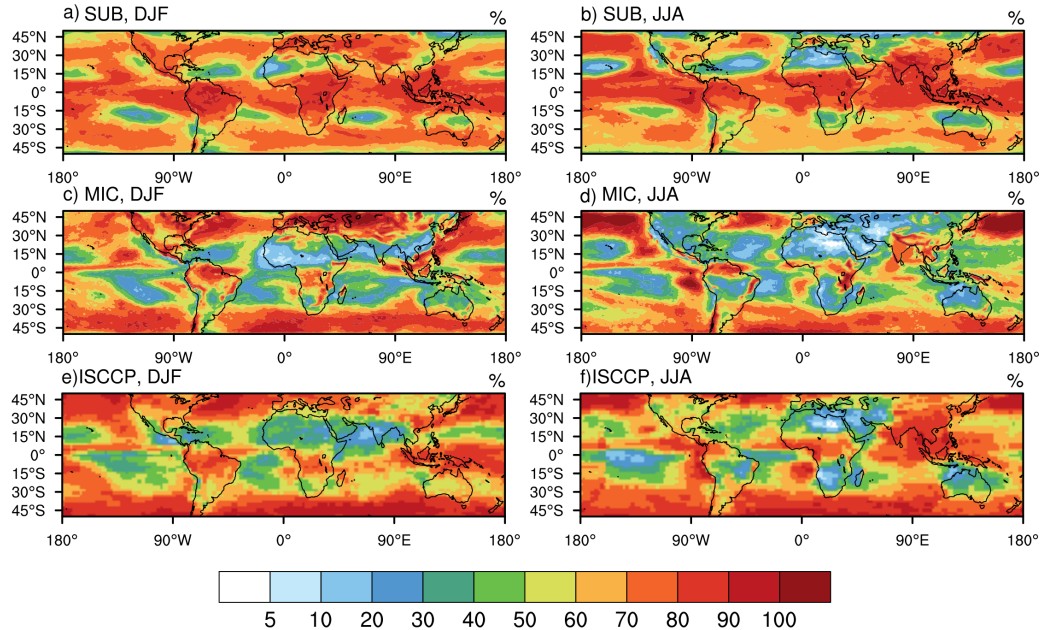

**Figure 3.** RegCM4 simulations using the ISCCP simulator using SUB scheme for DJF (a) and JJA, and using MIC scheme for DJF (c) and for JJA (d). Satellite observations of ISCCP total cloud amounts (unit in %) for JJA (a) and DJF (c).

**Table 2.** Global means (and spatial correlations with observations) of total cloud fractions for JJA and DJF 2007 of SUB and MIC versus observations.

| Fields | RegCM4 (SUB) + ISCCP Simulator Mean (Corr) | RegCM4 (MIC) + ISCCP Simulator Mean (Corr) | Obs: ISCCP Mean |
|---|---|---|---|
| Total Cloud Fraction JJA (%) | 67.35 (0.39) | 60.04 (0.69) | 64.66 |
| Total Cloud Fraction DJF (%) | 68.44 (0.32) | 61.52 (0.67) | 66.07 |

detailed investigation of the model clouds representation, we calculated the contributions from the
high (50-440 hPa, mainly cirrus and deep cumulus clouds), mid (440-680 hPa) and low (> 680 hPa, mainly shallow cumulus and stratocumulus) level clouds and compared them with estimates from the Cloud-Aerosol Lidar and Infrared Pathfinder Satellite Observations (CALIPSO, Winker et al. 2010) data. These are shown for JJA (results are similar for DJF) in Figures 4. The GCM-Oriented CALIPSO Cloud Product GOCCP data (Chepfer et al., 2010), 2°x 2°, are used for the model eval-
uation as they are designed for comparisons with output from the CALIPSO satellite simulator. In the observations high clouds occur along the ITCZ, and especially over the tropical continental areas, and over the midlatitudes storm track regions. Mid-level clouds are also prominent in the storm track regions and some tropical areas, while low clouds, including shallow cumulus and stratiform clouds, are prevalent over cooler subtropical oceans in correspondence of the descending branch of

the Hadley Cell. Both model versions capture rather well the distribution of low clouds, except over the Southern oceans, where only the MIC simulated some shallow stratiform cloud cover. The low level cloud cover averaged over the domain is essentially the same in the two schemes (Table 3), and slightly lower than the CALIPSO product. The largest differences between the two schemes occurs in the simulation of high and medium level clouds. Compared to the SUB scheme, the MIC produces much lower values of high clouds ( 25% vs. 64% for the domain average) and greater values of mid-level clouds ( 11% vs. 7%), in both cases considerably increasing the agreement with the CALIPSO data. A possible explanation could be related to the different approach in treating the convective detrainment: while in MIC the detrainment produced by the convection scheme is given as an input to the microphysics scheme and is therefore subjected to microphysical processes, in SUB the detrainment is a source of cloud liquid water and is not involved in the formation of rain until the following time step. Another possibility is that the SUB scheme does not include ice physics, which would be dominant at high altitudes. For example, ice crystals tend to aggregate faster than liquid droplets and thus precipitate more efficiently to lower levels. Note that the difference of the results between the assessments with the ISCCP (Figure 3) and CALIPSO (Figure 4) data suggests that the SUB scheme tends to overestimate optically thin clouds not detected by ISCCP. In fact ISCCP is able to detect clouds with optical depths greater than 0.15-0.25 (over ocean and land), while CALIPSO can measure optically thinner clouds with depths greater than 0.03 Rossow et al. (1996).

An even more accurate analysis of cloud vertical distribution can be carried out with the use of

**Table 3.** Global means of high, medium and low clouds of SUB and MIC versus observations.

| Fields | RegCM4 (SUB) + CALIPSO Simulator Mean | RegCM4 (MIC) + CALIPSO Simulator Mean | Obs: CALIPSO Mean |
|---|---|---|---|
| High Clouds (50-440 hPa) (%) | 64.33 | 24.85 | 31.97 |
| Medium Clouds (440-680 hPa) (%) | 6.62 | 11.10 | 16.53 |
| Low Clouds ( >680 hPa) (%) | 29.22 | 29.10 | 35.59 |

the Multi-angle Imaging SpectroRadiometer MISR (Muller et al. 2002) data. MISR uses nine cameras providing images with approximately 275 m sampling in four narrow spectral bands, spanning much of the angle range over which cloud reflectivity varies. This leads to a more accurate retrieval of albedo than the use of a single camera. Naud et al. (2002), however, found that in the case of multi layered clouds MISR often "sees" through the thin upper level clouds and mostly refers to low level cloud layers. The MISR retrievals can be processed to produce joint histograms of Cloud Top Height (CTH) and Optical Depth (OD) used specifically for a comparison with the COSP output and available on the CFMIP observational dataset website. To compare with the MISR retrievals, we postprocessed the RegCM4 data with the MISR simulator described in Marchand et al. (2010). An even more accurate analysis of cloud vertical distribution can be carried out with the use of the

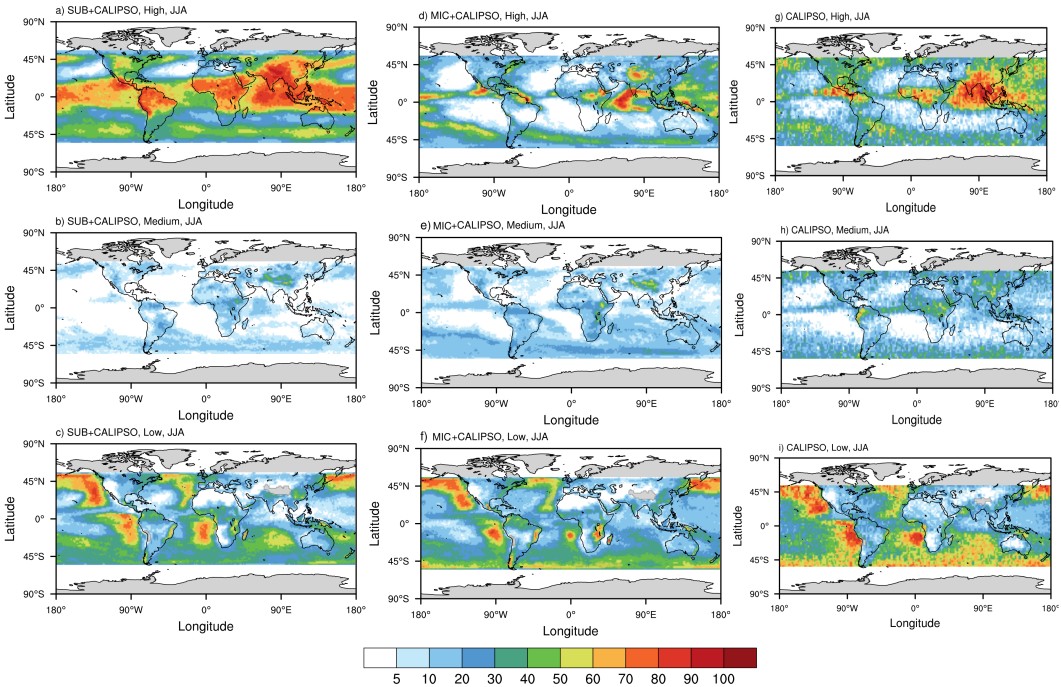

**Figure 4.** Left panels show RegCM4's high, middle, and low clouds (from top to bottom) using SUB and CALIPSO simulator for during JJA 2007 (unit in %). Middle panels shows the same fields using MIC and right panels show CALIPSO observations.

Multi-angle Imaging SpectroRadiometer MISR (Muller et al. 2002) data. MISR uses nine cameras, each of which makes images with approximately 275 m sampling in four narrow spectral bands, spanning much of the range of angles over which cloud reflectivity varies, thereby leading to a more accurate retrieval of albedo than the use of a single camera. Naud et al. (2002), however, found that in the case of multi-layered clouds MISR often through the thin upper level clouds and mostly refers to low level clouds layers. The MISR retrievals can be processed to produce joint histograms of Cloud Top Height (CTH) and Optical Depth (OD) used specifically for a comparison with COSP output and available on the CFMIP observational dataset website. To compare with the MISR retrievals, we postprocessed the RegCM4 data with the MISR simulator described in Marchand et al. (2010). Figure 5 reports the MISR histograms of optical depth vs. cloud top height averaged over the tropical band domain. It shows a bimodal distribution of cloud elevations, with two maxima in cloud fractions. One occurs at low altitudes, between 0 and 2.5 km, across a range of optical depths, from 0.8 to 16.2. The second is found at higher levels, between 5 and 9 km with optical depths of 2.45-16.2. Postprocessing the SUB output with the MISR simulator confirms an overestimation of high, thin clouds with optical depths lower than 2.45, along with thicker clouds with optical depth higher than 16.20. The MIC postprocessed output tends to underestimate low optical depth clouds ($\tau$<16.2) and to overestimate high clouds with optical depths greater than 41.5. As already found

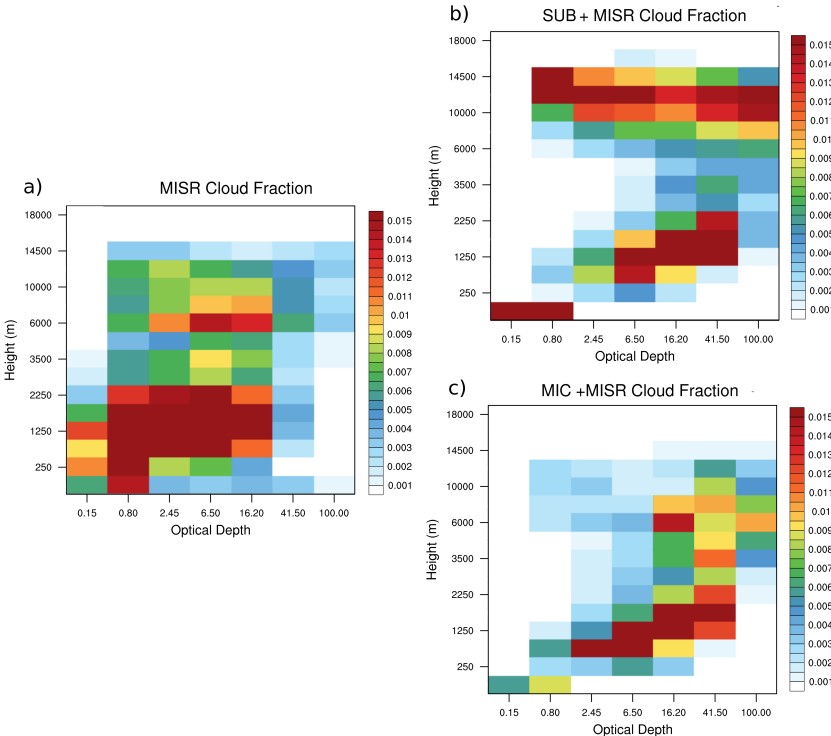

**Figure 5.** a) Joint histograms of cloud top height and optical depth for MISR observations for JJA. b) Joint histograms of cloud top height and optical depth for SUB using MISR simulator for JJA. c) Joint histograms of cloud top height and optical depth for MIC using MISR simulator for JJA. The colour scale represents the cloud fraction in adimensional units, from 0 to 1

using the CALIPSO simulator the main differences between the two schemes occur in the simulation of high clouds. Both schemes show a tendency to underestimate thin low clouds with optical depths lower than 2.45, although the MIC's low clouds exhibit a wider range of optical depths more in line with observations. While an underestimation of low clouds is a common problem in climate models (e.g. Nam et al., 2012; Zhang et al., 2005) a reason for the overestimation of thick clouds found here may reside in the fact that even if in real systems only part of a 100 km grid area experiences a strong upward motion, the mean vertical velocity for the whole model grid box is upward, leading to an updraft for the entire gridbox. A reason for the overestimation of low optically thick clouds can be related to the coarse horizontal model resolution (100 km) which does not resolve surface-heterogeneity, topography and shallow mesoscale circulations. Future studies will evaluate the model performance at higher horizontal resolutions.

### 3.3 TOA cloud radiative forcing

In this section we assess the cloud influence on the model radiation budget via an analysis of the CRF (Ramanathan et al., 1989), defined for the shortwave (SW) and longwave (LW) spectra as:

$$CRF = F^{cld} - F^{clr} \tag{18}$$

where $F$ is the net downward (i.e. downward minus upward) shortwave (SW) or longwave (LW) flux, the index $clr$ designates clear sky and $cld$ designate all-sky conditions. The CRF is calculated at the top of the atmosphere (TOA) for which observations are more reliable. The simulated CRFs

are compared to the corresponding fluxes from the Clouds and the Earth's Radiant Energy System CERES ERBA-like Monthly Geographical Averages (ES-4) observations (Wielicki, 2011), with an horizontal resolution of $2.5°$x $2.5°$. Figure 6 shows the TOA $CRF_{LW}$, where the values are positive

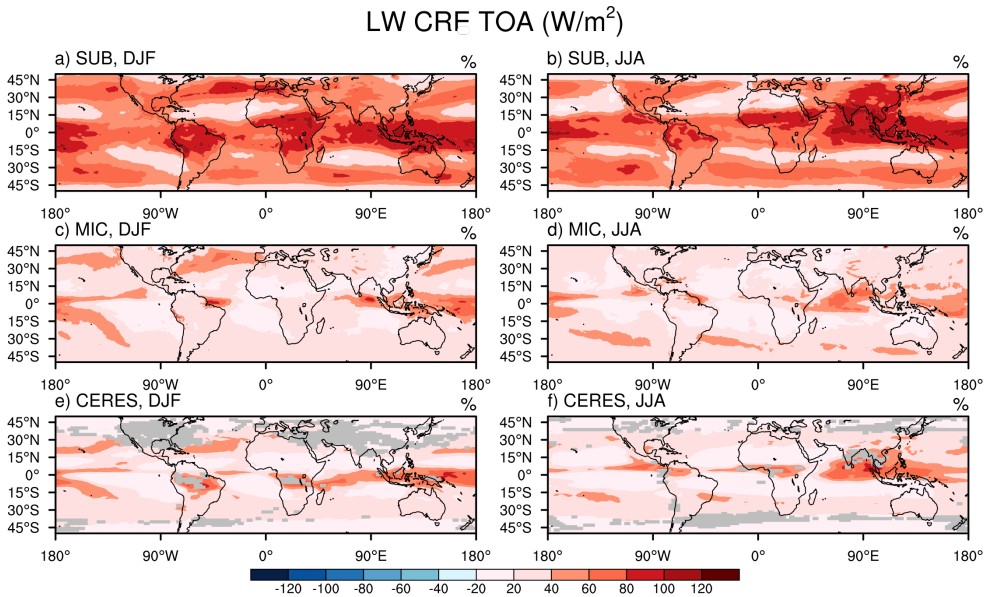

**Figure 6.** Simulated 10-yr mean TOA LW radiation budget for DJF (left panels) and JJA (right panels) by SUB and MIC and CERES observations. Grey areas indicate missing values.

because the net upward TOA LW flux is greater with clear skies than with cloudy skies due to the relatively low temperatures of clouds. The figure indicates that MIC matches observations much

better than SUB. The $CRF_{LW}$ biases in SUB and MIC simulations are consistent with those of the cloud fraction: with MIC the model simulates a smaller $CRF_{LW}$ because its clouds are lower and less extensive than with SUB, where the large overestimate of high clouds reduces excessively the infrared cooling to space. Overall the SUB scheme overestimates the domain-average $CRF_{LW}$ by 38 W m$^{-2}$ while the MIC is much closer to observations, with a bias of 8 W m$^{-2}$ (see Table 4).

Figure 7 shows the simulated and observed $CRF_{SW}$. In this case the values are negative because the net shortwave flux (defined as positive in the downward direction) for cloudy skies is smaller than for clear skies due to the cloud reflectivity. The excessive upper level cloud cover in the SUB run

yields too much SW reflection and therefore the domain-average SW values are about 40 W m$^{-2}$ lower than observed. The MIC scheme, by reducing the upper level cloud cover, reduces the upward SW flux and therefore yields values closer to observations (domain average bias of 10 W m$^{-2}$). However a substantial bias still persists in areas where both high and low clouds are well represented (e.g. around 45°S). This bias can be attributed to the underestimation of thin low clouds as shown by the MISR simulator analysis (Figure 5). Even if the overestimation of low cloud reflectivity is a common problem for many GCMs (e.g. Nam et al., 2012; Zhang et al., 2005) a reason for our overestimation of low optically thick clouds can be related to the coarse horizontal resolution (100 km) which does not resolve surface-heterogeneity, topography and shallow mesoscale circulations. Future studies will evaluate the model performance at higher horizontal resolutions. When looking at the full CRF, i.e. the sum of CRF$_{SW}$ and CRF$_{LW}$ (Figure 8 and Table 4), we see that essentially the model biases tend to compensate, yielding values close to each other for the two schemes and not far from observations (although on a domain average the MIC is still closer to observations by a few W/m2). In some tropical monsoon regions the longwave gain in the SUB scheme appears to be larger than the shortwave loss, leading to an overall heating which is less pronounced in the MIC scheme. To summarize the findings of this section, the new cloud parameterization has a strong

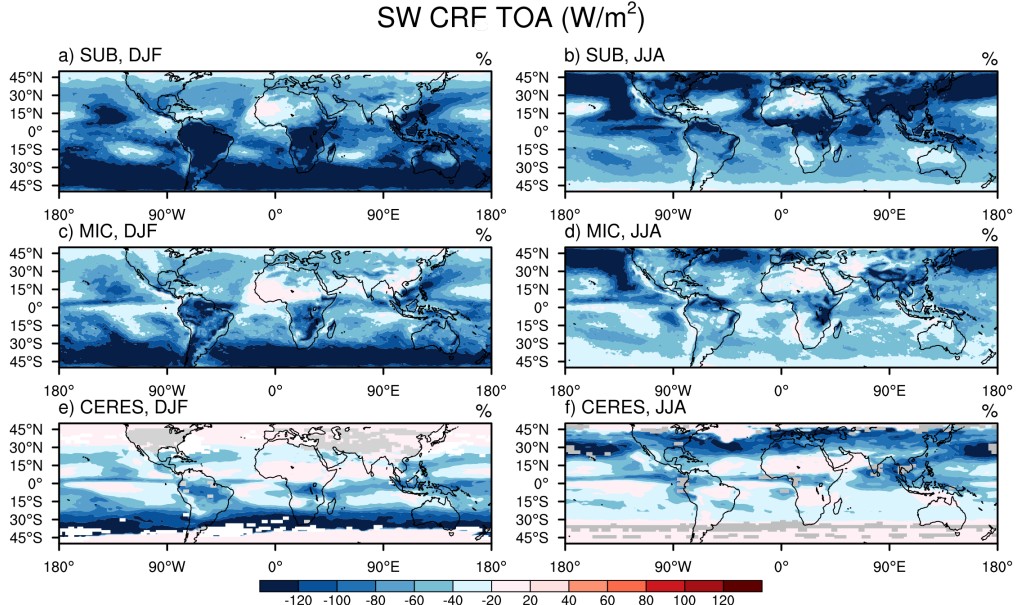

**Figure 7.** Simulated 10-yr mean TOA SW radiation budget for DJF (left panels) and JJA (right panels) by SUB and MIC and CERES observations. Grey areas indicate missing values.

effect (leading to closer agreement with observations) on the partitioning of the CRF in its shortwave and longwave components, although the total cloud forcing is similar to that of the old scheme due

Total CRF TOA (W/m$^2$)

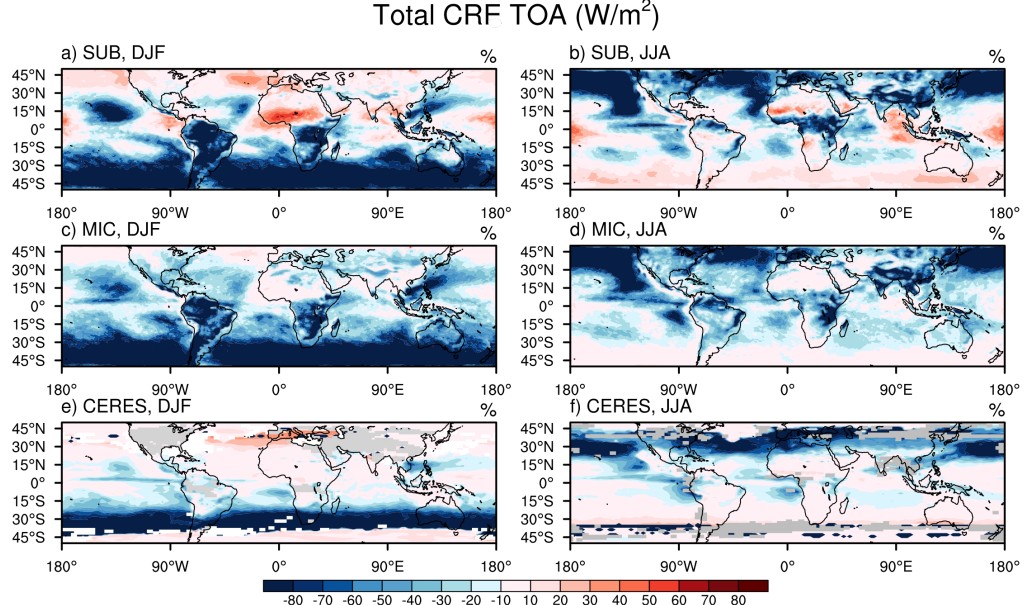

**Figure 8.** Simulated 10-yr mean net TOA CRF for DJF (left panels) and JJA (right panels) by SUB and MIC and CERES observations. Grey areas indicate missing values.

**Table 4.** Global means of MIC and SUB radiation fields versus observations.

| Fields | RegCM4: MIC | RegCM4: SUB | Obs: CERES |
|--------|-------------|-------------|------------|
| TOA CRF$_{LW}$ JJA (W m$^{-2}$) | 28.8 | 58.3 | 20.6 |
| TOA CRF$_{LW}$ DJF (W m$^{-2}$) | 29.9 | 59.6 | 21.2 |
| TOA CRF$_{SW}$ JJA (W m$^{-2}$) | -50.1 | -82.4 | -40.8 |
| TOA CRF$_{LW}$ DJF (W m$^{-2}$) | -53.4 | -85.3 | -40.6 |
| TOA CRF$_{tot}$ JJA (W m$^{-2}$) | -21.3 | -24.1 | -20.2 |
| TOA CRF$_{tot}$ DJF (W m$^{-2}$) | -23.5 | -25.7 | -19.3 |

to cancellation of biases. This is mostly attributed to the reduction of high level clouds found in the previous section.

## 4 Summary and conclusion

We here present the new resolved cloud microphysical parameterization implemented in the Regional Climate Model RegCM4 and some of the improvements that the new scheme brings to the model. To test the scheme we intercompared two 10-yr simulations using the RegCM4 with and without the new scheme over a tropical band domain. Our main results can be summarized as follows.

1. The new microphysics scheme (MIC) did not have a strong effect on simulated precipitation, although, compared to the original SUBEX scheme (SUB) it generally reduced precipitation amounts over land and increased them over ocean. In some cases this lead to a better agreement with observations while in others it worsened this agreement. In view of the model sensitivity to different precipitation schemes and of the uncertainty in precipitation observation products, an unambiguous assessment of the effect of the new scheme on the model performance in simulating precipitation requires large ensembles of model simulations and is left for future work.

2. Conversely, the new scheme had a strong effect on the simulation of cloudiness, and in particular it produced to a decrease in simulated upper level thin cirrus clouds, which increased agreement with observations and lead to an amelioration of a long-standing problem in the RegCM system (e.g. Giorgi et al. 1999). In general, the new scheme improved the vertical cloud profile in the model.

3. Despite having a small effect on the total CRF, the new scheme considerably improved its partitioning into longwave and shortwave components. This is mostly because of the reduction of the upper level cloud bias in the original scheme noted above.

The preliminary tests described here of the new microphysics scheme introduced in RegCM4 provide encouraging indications of it usefulness in improving the description of precipitation and especially cloud processes in the model. In particular, the fact that the main effect of the scheme is found in the simulation of high level clouds suggests that the inclusion of ice physics plays an important role in improving the model performance. More comprehensive sets of experiments are obviously needed in order to test the scheme in different model settings, especially towards its use in very high resolution, convection permitting simulations. We also need to assess the scheme's sensitivity to the use of different physics options in the model, particularly convection. All these issues are left for future work within the user community of the RegCM4 system. We also stress how the implementation of the COSP post processing program within the RegCM4 framework represents a new important tool for future research on the model representation of clouds and the hydrologic cycle.

## 5 Code availability

The code is available under GPL v2 license as part of the RegCM codebase from version 4.4 onward from the ICTP gforge website:

http://gforge.ictp.it/gf/project/regcm/frs.

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
