# Peer review of "Numerical framework and performance of the new multiple phase cloud microphysics scheme in RegCM4.5: precipitation, cloud microphysics and cloud radiative effects."

_Geoscientific Model Development, 2016_

## Referee Comment (RC1) · Anonymous Referee #1 · 5 Apr 2016

The paper describes the implementation of an improved cloud microphysics scheme for stratiform clouds within the RegCM4.5 model. The scheme introduces a prognostic representation of cloud water, ice, rain and snow in the model improving the physical basis for simulating mixed phase clouds and microphysical processes. The performance of the model is evaluated using the COSP simulator and comparing cloud radiative forcing to observational estimates. The paper is interesting and well written and requires only few changes. Since a few pieces of information are missing in the paper, prohibiting a comprehensive understanding of the results, I recommend major revisions.

Main comments:

1. It is not clear to me whether you have tuned the model after introducing the changes to the microphysical scheme. I assume that the original model was tuned with the SUBEX scheme to reproduce the radiative budget within the area covered. Could it be that the tuning forces the model to simulate a high amount of high clouds to balance a spurious heating in the model? And, in case you did not tune the model with the new microphysics scheme, you may allow the model to simulate a more realistic cloud field? Please give details on the model tuning and its implication for the results.

2. Why do you use random overlap? Most large scale models use maximum random overlap. Of course it depends on the layer thickness which overlap is more appropriate. How many of your 23 layers are in the troposphere and what is the resulting vertical resolution in the troposphere?

3. The differences between dX/dt and deltaX/delta t ....don't seem to be defined clearly enough – see comments further down (5. and 8.).

Minor comments:

1. You may want to give the full name for the SUBEX scheme when it is first mentioned.

2. Figure 1: What is the process that converts rain into cloud liquid or snow into cloud ice? You probably want to get rid of the arrow head pointing up. The arrow pointing from snow to water vapor should say sublimation and only point up. The evaporation arrow should only point up.

3. Equation 2: the sums should go from y=1 up to m and not from x=1.

4. Text before the equation after equation 2 (which is not equation 3!): You say it is an nxn matrix. But n is your time step! If you want to be consistent with equation 2 it should be an mxm matrix. It should also say 'm = 3 category system' instead of 'n=3'.

5. Equation 4: L should have an index x instead of x being in brackets. I think it

GMDD
should be dT/dt instead of deltaT/deltat? Why do you subtract the source of  $q_x$  due to convective outflow and due to sedimentation from  $dq_x/dt$ ? These sources of water/ice should be a sink in temperature in the same way as  $dq_x/dt$ . If there is a source of  $q_x$  due to sedimentation and convective detrainment, then  $dT_L/dt$  should not be =0.

6. Equation 5: Please give values/expressions for p, alpha and gamma.

7. Line 157-160: Not only do the time scales need to be fast but the ice crystal number needs to be high as well.

8. Equation 6: If the left hand side includes large scale advection already, then it is not clear to me why there is a second term on the right hand side.

9. Equation 8: this equation together with the diagnostic cloud scheme removes any supersaturation relative to ice. In line 177 you say 'condensation is a source of ice as homogeneous freezing takes place.' – it should say 'deposition' and the remainder of the sentence should be reformulated explaining that homogeneous freezing would only take place at high ice supersaturations but here in connection with the diagnostic cloud scheme deposition is handled just as condensation removing any supersaturation instantaneously.

10. Equation 11: You talk about evaporation due to turbulent mixing but you do not mention that by not resolving ice supersaturation you neglect the fact that there could be also an increase in ice mass due to turbulence.

11. Equation 12: The value of alpha does not seem to matter. The source term for any  $q_x$  is here D.

12. Equation 15: What are the values of b\_1, P\_loc and c\_0 ?

13. Later on in the text you do not mention which autoconversion formulation you use. It is not clear to me why you need all 4 alternative formulations here in the paper if you (presumably) only use one.
14. Table 1: You probably want to list the ISCCP simulator in the last 2 lines as well.

15. Line 299: Please complete the information for this citation in the publication list.

16. Table 2: How large is the interannual variability in global mean total cloud fraction? Are the differences of the simulated coverage significantly different from the observations or could the simulations be a member of the distribution of observed cloud coverages? I assume you have quite a few years of data from the ISCCP observations and could easily check this. Similarly in table 4 you should be able to give an estimate for the variability of CRF. I assume that the differences in SW and LW fluxes are huge compared to the interannual variability but for CRF\_tot it is not that obvious anylonger. Please note that the fourth row should say TOA CRF\_SW ..... and not LW.

17. Figure 4e should say MIC Medium JJA

18. Figure 7: In the CERES data, is the white in the very south in DJF and the very north in JJA missing values or really values close to zero.

GMDD

---

## Referee Comment (RC2) · Anonymous Referee #2 · 11 Apr 2016

**1. Decision**

The authors introduce a more sophisticated and complex cloud parameterization in the regional model RegCM4, which allows a more realistic representation of the microphysics processes than the standard scheme. After describing the new scheme, they evaluate the representation of precipitation, clouds, and TOA radiation against observations for the new and old scheme, using satellite simulators for the cloud part. While the improvement of precipitation is unclear, the representation of clouds is clearly better using the new cloud scheme, particularly in the upper levels, leading to an improvement of the simulation of radiation at TOA.

This paper is perfectly within the scope of the journal and well written as well as clearly presented. The topic is of particular interest as it shows how a more physical representation of microphysics in models can lead to a better representation of cloud/radiation when compared to observations. Besides, the authors show once again how a multi-observational dataset approach help understanding the model's biases, while using a consistent and solid method to compare model and obs via the simulators.

However, the problematic in the introduction could be substantially improved and some information regarding the observations in the manuscript is missing. In addition, the authors failed to explain the reasons behind some of their results, which make me think the paper need a major revision before being published.

My detailed comments are listed hereafter.

**2. Main concerns:**

1) Although the authors used the COSP package, they didn't describe which version of the package is used. Depending on the version, they might have used the new CALIPSO cloud phase diagnosis (ver 1.4), which allows distinguishing ice clouds from liquid clouds. This would have been particularly interesting in this study, i.e. the ice-to-liquid ratio vs. T or z.

Even though, the COSP version used here is anterior to 1.4, the authors should consider using the vertically resolved cloud fraction of CALIPSO to assess their model. It would give us more information about how the model represents the vertical structure of clouds (better than only 3 vertical layers, low mid and high).

2) The introduction misses some important references to stress the importance of having a more realistic representation of microphysics in climate models and what has been already done in the field as well as in the observations. For example, how cloud phase determination affects the GCMs/RCMs, does it really matter? Cesana et al (2015) (also Komurcu et al (2014)) showed that the climate models particularly under-estimate the super-cooled liquid clouds compared to observations; and a more complex microphysics helps reducing the problem. Tan et al (2016) recently showed that better representing those supercooled liquid clouds (constrained using CALIPSO) might drastically change the equilibrium climate sensitivity of climate models. Moreover, there has been a lot of work on the observed cloud phase that is not mentioned here. It could be helpful for the reader to know that. For example, liquid and ice particles may co-exist for hours (Korolev et al., 2003) and sometimes during days (de Boer et al., 2009). Also, observations showed substantial presence of supercoold liquid at temperature as low as -35°C, in agreement with insitu observations (Cesana et al., 2016).

3) In some part of the manuscript, the authors do not explain the reason of the simulated bias. I think of the low cloud problem, which affects the TOA radiation in Sect. 3.2 and 3.3

4) Finally, not enough details are given regarding the observations used in the manuscript. The authors should mention where they got it and what is the resolution and time period they used.

**3. Minor comments:** Abstract**

Line 5: five

Line 8-10: A little bit confusing as not 10-year are used for COSP comparison.

Also, I would not say the COSP simulator but either satellite simulators or the full definition the cloud feedback... package.

**Introduction**

Line 54: Please define COSP.

**Section 2.**

The authors should consider doing a small summary of the new scheme at the beginning of Sect. 2.2 as it is done for the old scheme in Sect. 2.1. It would highly help readers not expert in model development and readers in general to identify the main changes.

Line 256: 5  $\rightarrow$  five

Sect. 2.2.2: The authors state that one year of simulation might be enough to draw solid conclusions, which I also believe. To strengthen this statement, though, the authors might use CMIP5 model outputs that use the same core as RegCM4 (e.g. ecearth) and show that the inter-annual variation of COSP fields is smaller than the model-obs bias.

Fig. 2 is very difficult to read in its present form. The authors should consider adding either the bias compared to observations or may be just the difference between the two experiments to help the reader locate the differences.

Moreover, no explanation is given for these differences. Could the authors at least guess the main reason for this slight improvement (line 296)?

Time period and resolution of the obs?

**Section 3**

Did the authors use the cfmip-obs ISCCP dataset, which are designed to be consistent with the simulator? If so, please mention it and refer to the website.

Tab. 2: It's a detail but SUB results should appear in the left column rather than the mid column to be consistent with the order to which it appears in the figures: SUB  $\rightarrow$  MIC $\rightarrow$  OBS

Figure 3: Again, a difference and/or a bias plot in Fig. 3 might help identifying the improvements.

A correlation between obs and simulation could be added to tab 2 and I bet it would be higher for the MIC scheme, highlighting the fact that even though the mean is worse in MIC than in SUB, using MIC scheme improves the distribution of clouds in the model.

Line 325: The authors might add "GCM-oriented" CALIPSO estimates to be more specific.

Line 326: Please use a more recent reference for CALIPSO: Winker et al., 2010, doi:10.1175/2010BAMS3009.1

Line 327: Which version of CALIPSO-GOCCP did you use and what about the resolution and the time period? Judging from the figure, it seems to be only one season and 1degx1deg grid. I would strongly encourage the author to at least pick the 2x2deg grid and averaged over all available seasons to smooth the noise. As mentioned before, the inter-annual variation is lower than the mod-to-obs bias anyway and should change the pattern of the bias. The 1x1deg grid is also very noisy because of the poor overlap due to CALIPSO polar orbit.

Line 346: liquid droplets rather than cloud droplets.

Table 3: same as for Tab 2, SUB  $\rightarrow$  MIC  $\rightarrow$  OBS and maybe the authors should adding correlation numbers.

For MISR, same questions as for ISCCP and GOCCP, are these from CFMIP-obs? And what is the resolution?

Line 367-369: May these low clouds be the shallow cumulus cloud, implying that the RegCM4 model struggle to represent the transition from strato to shallow cumulus clouds, as many other models? Besides, it is in agreement with the few too bright problem, too few low clouds but optically too thick (e.g. Nam et al., 2012, doi:10.1029/2012GL053421)

**Section 3.3**

Did the authors use the CERES-EBAF data, specially designed for model evaluation? Please, clarify and define the resolution and time period.

While the explanation for the CRFlw is straightforward, the upper cloud issue does not explain all of the CRFsw bias, and the authors do not refer to the other reasons of the bias. As mentioned before, the low clouds have been shown to be mostly too reflective in many GCMs for quite a while now (e.g. Nam et al., 2012; Zhang et al., 2005). It seems to be also the case for RegCM4. The large bias in the CRFsw remains even in region where the upper and lower clouds are well reproduced by the model (e.g. around 45°S). Could this be because of the thin low clouds missed by your model as shown by the MISR simulator analysis? Could you i) locate these thin low clouds on a map and ii) propose an explanation of why they are so optically thick?

Line 418: cirrus instead of stratocumulus

**4. References**

- Cesana G., H. Chepfer, D. Winker, X. Cai, B. Getzewich, H. Okamoto, Y. Hagihara, O. Jourdan, G. Mioche, V. Noel and M. Reverdy, 2016: Using in-situ airborne measurements to evaluate three cloud phase products derived from CALIPSO, J. Geophys. Res., Accepted.
- Cesana, G., D. E. Waliser, X. Jiang, and J.-L. F. Li, 2015: Multimodel evaluation of cloud phase transition using satellite and reanalysis data, J. Geophys. Res. Atmos., doi:10.1002/2014JD022932
- de Boer, G., Eloranta, E. W. and M. D. Shupe (2009), Arctic Mixed-Phase Stratiform Cloud Properties from Multiple Years of Surface-Based Measurements at Two High-Latitude Locations, J. Atmos. Sci., 66:9, 2874-2887, doi: 10.1175/2009JAS3029.1
- Komurcu, M., T. Storelvmo, I. Tan, U. Lohmann, Y. Yun, J. E. Penner, Y. Wang, X. Liu, and T. Takemura (2014), Intercomparison of the cloud water phase among global climate models, J. Geophys. Res. Atmos., 119, doi:10.1002/2013JD021119.
- Korolev, A. V., Isaac, G. A., Cober, S. G., Strapp, J. W. and Hallett, J. (2003), Microphysical characterization of mixed-phase clouds. Q.J.R. Meteorol. Soc., 129: 39–65. doi: 10.1256/qj.01.204
- Nam C., S. Bony, JL Dufresne, H. Chepfer, 2012: The 'too few, too bright' tropical lowcloud problem in CMIP5 models, Geophys. Res. Lett., 39, 21, idoi:10.1029/2012GL053421.
- Tan, I. et al., 2016: Observational constraints on mixed-phase clouds imply higher climate sensitivity, 10.1126/science.aad5300
- Winker, D. M., J. P., J. A. Coakley Jr., S. A. Ackerman, R. J. Charlson, P. R. Colarco, P. Flamant, Q. Fu, R. M. Hoff, C. Kittaka, T. L. Kubar, H. Le Treut, M. P. McCormick, G. Mégie, L. Poole, K. Powell, C. Trepte, M. A. Vaughan, and B. A. Wielicki, (2010), The CALIPSO Mission: A Global 3D View of Aerosols and Clouds. Bull. Amer. Meteor. Soc., 91, 1211–1229, doi:10.1175/2010BAMS3009.1
- Zhang, M., et al. (2005), Comparing clouds and their seasonal variations in 10 atmospheric general circulation models with satellite measurements, J. Geophys. Res., 110, D15S02, doi:10.1029/2004JD005021.

---

## Referee Comment (RC3) · Anonymous Referee #3 · 11 May 2016

The paper introduces a needed update to the moist physics in the RegCM4 community regional climate model, namely the inclusion of ice phase microphysics. Given the wide use of RegCM4 it is likely that this paper will be heavily referenced. The paper is well written and there are only a few minor changes needed to clarify and strengthen it.

1. There are many microphysical schemes in existence, some of which are more detailed than the scheme here and some less. It would be appropriate to discuss briefly the rationale for choosing this particular scheme for inclusion in RegCM4 compared to other options.

[Figure]

2. At line 68: Is there no rainwater evaporation in SUBEX?

3. Are any of the parameters in the new scheme known or suspected to be sensitive to grid spacing? Intuitively it would seem that some of the parameters (such as those in Equation 5) should approach limiting values for very small grid volumes and as such their most appropriate values could vary with grid spacing.

4. Line 195, "condensate" should be "condense."

5. The RHS of equation (12) simply works out to D, since alpha + (1-alpha) = 1. This does not seem correct. Are there missing subscripts or other corrections needed?

6. Line 205, regarding the four different autoconversion parameterizations: Are these user-selectable, or are different parameterizations invoked automatically by the scheme depending on the physical conditions?

7. Equation (14), the species for which ql and qcrit apply should be clarified. Typically the rate on the LHS applies to precipitation and the humidity on the right-hand side is cloud water, but this equation has ql on both the LHS and RHS implying a positive feedback (which seems unusual).

8. Line 244, The reference on IFS documentation does not appear in the list of references, or at least not under that title. Please give sufficient bibliographic information so that the reader can access this document.
* * *

---

## Author Comment (AC1) · 7 Jun 2016

First, we would like to thank all the reviewers for their careful reviews and constructive comments, which helped to improve the quality and clarity of the paper.

Anonymous Referee #1

The paper describes the implementation of an improved cloud microphysics scheme for stratiform clouds within the RegCM4.5 model. The scheme introduces a prognostic representation of cloud water, ice, rain and snow in the model improving the physical basis for simulating mixed phase clouds and microphysical processes. The performance of the model is evaluated using the COSP simulator and comparing cloud radiative forcing to observational estimates. The paper is interesting and well written and requires only few changes. Since a few pieces of information are missing in the paper, prohibiting a comprehensive understanding of the results, I recommend major revisions.

Main comments: 1. It is not clear to me whether you have tuned the model after introducing the changes to the microphysical scheme. I assume that the original model was tuned with the SUBEX scheme to reproduce the radiative budget within the area covered. Could it be that the tuning forces the model to simulate a high amount of high clouds to balance a spurious heating in the model? And, in case you did not tune the model with the new microphysics scheme, you may allow the model to simulate a more realistic cloud field? Please give details on the model tuning and its implication for the results.

The model was tuned after introducing the new microphysical scheme, we wanted to test the best performance given by the new parameterization and afterwards compare it with the pre-existing scheme. The presence of an overestimation of high clouds using the SUBEX scheme, however, was not very sensitive to the SUBEX parameters, therefore clouds pattern wouldn't be much different by tuning the model before introducing the new scheme. The problem of excessive high level cloudiness has been a long-standing one within the RegCM system.

2. Why do you use random overlap? Most large scale models use maximum random overlap. Of course it depends on the layer thickness which overlap is more appropriate. How many of your 23 layers are in the troposphere and what is the resulting vertical resolution in the troposphere?

We thank the reviewer to point this out: we have actually used the max-random overlap assumption but there was a mistake in the text. Most of the 23 layers are located in the troposphere, as the model top is at about 50 hPa. In the mid-troposphere the vertical

resolution is about 0.75 sigma (i.e. about 75 hPa) , since higher level density is placed in the boundary layer.

3. The differences between dX/dt and deltaX/delta t : don't seem to be defined clearly enough – see comments further down (5. and 8.).

More clarifications about this in comments 5. and 8.

Minor comments:

1. You may want to give the full name for the SUBEX scheme when it is first mentioned.

Done.

2. Figure 1: What is the process that converts rain into cloud liquid or snow into cloud ice? You probably want to get rid of the arrow head pointing up. The arrow pointing from snow to water vapor should say sublimation and only point up. The evaporation arrow should only point up.

Done. This was a mistake.

3. Equation 2: the sums should go from y=1 up to m and not from x=1.

Done.

4. Text before the equation after equation 2 (which is not equation 3!): You say it is an nxn matrix. But n is your time step! If you want to be consistent with equation 2 it should be an mxm matrix. It should also say 'm = 3 category system' instead of 'n=3'.

Done.

5. Equation 4: L should have an index x instead of x being in brackets. I think it should be dT/dt instead of deltaT/deltat? Why do you subtract the source of q_x due to convective outflow and due to sedimentation from dq_x/dt? These sources of water/ice should be a sink in temperature in the same way as dq_x/dt? If there is a source of q_x due to sedimentation and convective detrainment, then dT_L/dt should not be =0.

We thank the reviewer for this comment as there was a mistake in eq. 4, now corrected with deltaT/deltat. We subtract the convective detrainment term and the advective flux since they do not represent changes in temperature due to the latent heating with changes of phase of water in the scheme itself. Recall that the T_L budget is just used over the cloud scheme, since the processes are solved both implicitly and explicitly. We thus need to use the conservation in T_L before and after the scheme to work out what T change is associated with the change in the family of qv, ql, and qi. Any "source" of microphysics variables that is *not* the result of phase change within the microphysics has to be accounted for, otherwise it will result in a superfluous change in temperature. Recall that the impact on condensation in the convective updraughts is already accounted for in the temperature budget of the convection scheme itself. We have modified the text to: "where Lx is the latent heat (of fusion or evaporation depending on the processes considered), Dqx is the convective detrainment and the third term in the brackets is the sedimentation term. We subtract the convective detrainment term Dqx and the advective flux terms to the rate of change of species qx (due to all the processes) because they represent a net TL flux not associated with latent heating with changes of phase of water in the scheme itself."

6. Equation 5: Please give values/expressions for p, alpha and gamma.

Done.

7. Line 157-160: Not only do the time scales need to be fast but the ice crystal number needs to be high as well.

Good point, although I would clarify the reviewers point in stating that it is not precisely a case of needing fast-timescale *and* high ice crystal number concentrations, since the former is a function of the latter. In the cases where homogeneous ice nucleation dominates, the assumption is very reasonable. However, even if the case of heterogeneous nucleation it is fairly reasonable, since in most updraughts, if the IN number concentration is low, homogeneous nucleation will kick in anyway. The limited

(albeit slow) depositional growth of the isolated IN prior to the homogeneous nucleation threshold being reached can be ignored at this level of approximation without impact. If IN are numerous enough to shut off homogeneous nucleation then the timescale is roughly on the same order as a GCM timestep and the assumption is still reasonable. We have added a statement to this effect.

Added in the text: "As stated by \cite{tompkins:07}, this is very good assumption if ice nucleation is predominately homogeneous in nature, although even if heterogeneous nucleation predominates it is still reasonable, since to cut off homonucleation completion IN concentrations need to be of an order of magnitude that results in the growth timescale is similar to a typical global model timestep (Kärcher and Lohmann 2003)"

8. Equation 6: If the left hand side includes large scale advection already, then it is not clear to me why there is a second term on the right hand side.

The equation is correct, the left hand side is the total derivative of the saturation mixing ratio (see IFS documentation). Please find all the steps of the equation in Figure 1.

9. Equation 8: this equation together with the diagnostic cloud scheme removes any supersaturation relative to ice. In line 177 you say 'condensation is a source of ice as homogeneous freezing takes place.' – it should say 'deposition' and the remainder of the sentence should be reformulated explaining that homogeneous freezing would only take place at high ice supersaturations but here in connection with the diagnostic cloud scheme deposition is handled just as condensation removing any supersaturation instantaneously.

We thank the reviewer for the comment and we have riformulated the sentence in this way: "and all the increase of cloud amount is a source of cloud water unless the process occurs within cold clouds, in which case deposition occurs and ice forms. Due to the diagnostic treatment of the cloud fraction, homogeneous freezing takes place and removes any supersaturation instantaneously."

10. Equation 11: You talk about evaporation due to turbulent mixing but you do not mention that by not resolving ice supersaturation you neglect the fact that there could be also an increase in ice mass due to turbulence.

The reviewer is correct and we were aware of this shortcoming already and now explicitly identify it as a caveat of this first implementation in the text.

Changed the text to: "A very simple treatment of turbulence mixing is adopted in this first version of the scheme that duplicates the approach of Tiedtke (1991) by treating turbulence as a sink of cloud water. As discussed by \cite{Tompkins:02} and \cite{Tompkins:05}, the sign of the turbulent impact on cloud water is only correct if the total water mixing ration $q_t= q_l+q_i$ is smaller than the saturation mixing ratio, otherwise mixing leads to an increase in cloud water. The intention is to correct this when a PDF-based cloud cover parametrization is later implemented."

11. Equation 12: The value of alpha does not seem to matter. The source term for any q_x is here D.

We thank the reviewer for the comment: there was a mistake in the equation, now corrected as follows:

$(\partial q_x)/\partial t=\alpha(T)D_x$

12. Equation 15: What are the values of b_1, P_loc and c_0 ?

We added the following text after equation (15): "where c0 = 1.67 10−4 s−1, b1 = 100 (kg m−2s−1) and Ploc is the local cloudy precipitation rate."

13. Later on in the text you do not mention which autoconversion formulation you use. It is not clear to me why you need all 4 alternative formulations here in the paper if you (presumably) only use one.

We wanted the paper to describe the scheme in all its features and options. Showing tests of all the autoconversion parameterizations was however beyond the scope of this

work. We added in the text: "Here, the default autoconversion parameterization is set to the Sundqvist's scheme (eq. 15) and sensitivity studies using different autoconversion schemes need to be carried out for specific applications."

14. Table 1: You probably want to list the ISCCP simulator in the last 2 lines as well.

Done.

15. Line 299: Please complete the information for this citation in the publication list.

Done.

16. Table 2: How large is the interannual variability in global mean total cloud fraction? Are the differences of the simulated coverage significantly different from the observations or could the simulations be a member of the distribution of observed cloud coverages? I assume you have quite a few years of data from the ISCCP observations and could easily check this. Similarly in table 4 you should be able to give an estimate for the variability of CRF. I assume that the differences in SW and LW fluxes are huge compared to the interannual variability but for CRF_tot it is not that obvious anylonger. Please note that the fourth row should say TOA CRF_SW : : :.. and not LW.

Thanks for pointing out this issue. Figure 2 compares the average of observations for the period of available data (1983-2007, panels g) and h) with the analogous fields for our analysis year (2007). It can be seen that 2007 is generally representative of the long term climatology, suggesting that the inter-annual variation in global mean high, medium and low total cloud fractions is not large and it is reasonable to choose only one season for detailed analysis. We have modified the fourth row.

17. Figure 4e should say MIC Medium JJA

Done.

18. Figure 7: In the CERES data, is the white in the very south in DJF and the very north in JJA missing values or really values close to zero.

We thank the reviewer for this comment. We re-made the three radiation plots with grey areas depicting missing values and added it in the caption.

[Figure]

$$\frac{dq_{sat}}{dt} = \frac{\partial q_{sat}}{\partial T}\frac{dT}{dt}$$

$$= \frac{\partial q_{sat}}{\partial T}\left[\frac{\partial T}{\partial t} + \omega\frac{\partial T}{\partial p}\right]$$

$$= \frac{\partial q_{sat}}{\partial T}\frac{\partial T}{\partial t} + \left[\frac{\partial q_{sat}}{\partial T}\omega\frac{\partial T}{\partial p}\right]$$

$$= \frac{\partial q_{sat}}{\partial T}\frac{\partial T}{\partial t}\bigg|_{diab} + \frac{\partial q_{sat}}{\partial p}\bigg|_{ma}\omega$$

**Fig. 1.** Passages of the equation for the total derivative of the saturation mixing ratio

[Figure]

[Figure]

**Fig. 2.** Same as Figure 3 of the paper but with two additional panels (g) and h)) with the average of total cloud for the period of avialble data (1983-2007).

---

## Author Comment (AC2) · 7 Jun 2016

First, we would like to thank all the reviewers for their careful reviews and constructive comments, which helped to improve the quality and clarity of the paper.

Anonymous Referee #2

The authors introduce a more sophisticated and complex cloud parameterization in the regional model RegCM4, which allows a more realistic representation of the microphysics processes than the standard scheme. After describing the new scheme, they evaluate the representation of precipitation, clouds, and TOA radiation against

observations for the new and old scheme, using satellite simulators for the cloud part. While the improvement of precipitation is unclear, the representation of clouds is clearly better using the new cloud scheme, particularly in the upper levels, leading to an improvement of the simulation of radiation at TOA. This paper is perfectly within the scope of the journal and well written as well as clearly presented. The topic is of particular interest as it shows how a more physical representation of microphysics in models can lead to a better representation of cloud/radiation when compared to observations. Besides, the authors show once again how a multi-observational dataset approach help understanding the model's biases, while using a consistent and solid method to compare model and obs via the simulators. However, the problematic in the introduction could be substantially improved and some information regarding the observations in the manuscript is missing. In addition, the authors failed to explain the reasons behind some of their results, which make me think the paper need a major revision before being published. My detailed comments are listed hereafter.

Main concerns:

1) Although the authors used the COSP package, they didn't describe which version of the package is used. Depending on the version, they might have used the new CALIPSO cloud phase diagnosis (ver 1.4), which allows distinguishing ice clouds from liquid clouds. This would have been particularly interesting in this study, i.e. the ice-to-liquid ratio vs. T or z. Even though, the COSP version used here is anterior to 1.4, the authors should consider using the vertically resolved cloud fraction of CALIPSO to assess their model. It would give us more information about how the model represents the vertical structure of clouds (better than only 3 vertical layers, low mid and high).

We added in the introduction: "...the new parameterization is also assessed using the recently available Cloud Feedback Model Intercomparison Project (CFMIP) Observational Simulator Package (COSP, version 1.3.2) (Bodas-Salcedo et al., 2011)". We have used COSP version 1.3.2 because the version 4 was not available when we started our work. We decided to choose the three layer cloud representation because
we wanted to compare our results with those by Franklin 2013 and because we were specifically interested in the representation of high vs. low clouds. Future studies will be carried out using the vertically resolved cloud fraction from CALIPSO and the other simulators (MODIS and CloudSat).

2) The introduction misses some important references to stress the importance of having a more realistic representation of microphysics in climate models and what has been already done in the field as well as in the observations. For example, how cloud phase determination affects the GCMs/RCMs, does it really matter? Cesana et al (2015) (also Komurcu et al (2014)) showed that the climate models particularly under-estimate the super-cooled liquid clouds compared to observations; and a more complex microphysics helps reducing the problem. Tan et al (2016) recently showed that better representing those supercooled liquid clouds (constrained using CALIPSO) might drastically change the equilibrium climate sensitivity of climate models. Moreover, there has been a lot of work on the observed cloud phase that is not mentioned here. It could be helpful for the reader to know that. For example, liquid and ice particles may co-exist for hours (Korolev et al., 2003) and sometimes during days (de Boer et al., 2009). Also, observations showed substantial presence of supercoold liquid at temperature as low as -35°C, in agreement with insitu observations (Cesana et al., 2016).

We modified the Introduction in the following way:

"Simpler microphysics schemes treat the cloud water prognostically and precipitating water diagnostically (e.g. Rotstayn, 1997; Pal et al., 2000). Observational data show that between -23 °C and 0 °C the occurrence of supercooled water is not negligible (Matveev, 1984), and liquid and ice particles can co-exist for hours and sometimes even days (e.g. Korolev et al., 2003; de Boer et al., 2009). Often cloud schemes diagnose the fraction of cloud water in the ice phase based on the local temperature (e.g. DelGenio et al., 1996). The diagnostic partitioning of cloud water into the liquid and ice phases assumes implicitly that processes within the cloud are fast compared to the

model time step, i.e. that the cloud variables are always in equilibrium. Therefore, a diagnostic representation is unable to describe the temporal variability and evolution of mixed-phase clouds and a prognostic treatment of cloud ice and water is necessary to represent the microphysical processes of the two phases (including their contrasting sedimentation rates). More complex microphysics schemes have been therefore introduced to treat separately the cold and warm cloud microphysics by solving prognostic equations for cloud liquid water and ice (e.g. Fowler et al., 1996; Lohmann and Roeckner, 1996). These schemes are especially important as climate models approach resolutions at which cloud physics processes, including convection, need to be explicitly described without the use of parameterization schemes (e.g. Prein et al. 2015). Recently, several studies have illustrated the importance of using a more realistic representation of cloud microphysics in climate models. For example, Cesana et al. (2015) and Komurcu et al. (2014) showed that climate models tend to underestimate the supercooled liquid clouds and models that prognose separately the liquid and ice mixing ratio give a better representation of cloud properties."

3) In some part of the manuscript, the authors do not explain the reason of the simulated bias. I think of the low cloud problem, which affects the TOA radiation in Sect. 3.2 and 3.3

Done (see the following points).

4) Finally, not enough details are given regarding the observations used in the manuscript. The authors should mention where they got it and what is the resolution and time period they used.

Done (see the following points).

Minor comments:

Abstract

Line 5-> five Done.

Line 8-10: A little bit confusing as not 10-year are used for COSP comparison. Also, I would not say the COSP simulator but either satellite simulators or the full definition the cloud feedback... package.

We modified the following text: "Various fields from a 10-yr-long integration of RegCM4 run in tropical band mode with the new scheme are compared with their counterparts using the previous cloud scheme and are evaluated against satellite observations. In addition, an assessment using the Cloud Feedback Model Intercomparison Project (CFMIP) Observational Simulator Package (COSP) for a 1-yr sub-period provides additional information for evaluating the cloud optical properties against satellite data." Introduction

Line 54: Please define COSP.

Done.

Section 2.

The authors should consider doing a small summary of the new scheme at the beginning of Sect. 2.2 as it is done for the old scheme in Sect. 2.1. It would highly help readers not expert in model development and readers in general to identify the main changes.

We think that the beginning of Section 2.2 already contains a summary of the scheme: we have modified the text at the beginning of Section 2.2 from "The model includes four hydrometeors..." to "The scheme includes four hydrometeors..." as we think this was misleading.

Line 256: 5 -> five

Done.

Sect. 2.2.2: The authors state that one year of simulation might be enough to draw solid conclusions, which I also believe. To strengthen this statement, though, the authors

might use CMIP5 model outputs that use the same core as RegCM4 (e.g. ecearth) and show that the inter-annual variation of COSP fields is smaller than the model-obs bias.

As we have already shown in the response to comment 16 of reviewer 1 and in the response to the comment for line 327 below, the clouds for the year we selected is quite representative of the long term average cloud cover in the ISCCP dataset. It is unclear to us what the suggestion by the reviewer is concerning the use of CMIP5 data, since the RegCM4 is quite different from all CMIP5 models so that comparison with the CMIP5 output would probably not help much in this regard. Finally, we note that in the literature the use of one year is considered to be sufficient for a first order evaluation of model clouds.

Section 3 Did the authors use the cfmip-obs ISCCP dataset, which are designed to be consistent with the simulator? If so, please mention it and refer to the website.

We modified the text in the following way: " The evaluation of total cloud cover is carried out using the GCM simulator-oriented International Satellite Cloud Climatology Project ISCCP cloud product (Pincus et al., 2012), which was prepared to facilitate the evaluation of the model simulated clouds within the framework of the Cloud Feedback Model Intercomparison Project (http://climserv.ipsl.polytechnique.fr/cfmip-obs). Data are averaged over the JJA and DJF 2007 seasons during the daytime, at a horizontal resolution of 2.5°x2.5°."

Tab. 2: It's a detail but SUB results should appear in the left column rather than the mid column to be consistent with the order to which it appears in the figures: SUB ! MIC!OBS

Done.

Figure 3: A difference and/or a bias plot in Fig. 3 might help identifying the improvements. A correlation between obs and simulation could be added to tab 2 and I bet it

[Figure]

would be higher for the MIC scheme, highlighting the fact that even though the mean is worse in MIC than in SUB, using MIC scheme improves the distribution of clouds in the model.

We added the correlation values in Table 2 and modified the text as follows: "In general, both schemes capture the horizontal distribution of clouds over the band domain in both seasons, with maximum cloud cover over the ITCZ and the mid-latitude storm track regions of both hemispheres. However an analysis of the spatial correlation between the two schemes and the observations reveals that the new parameterization improves the horizontal distribution of clouds (Table \ref{tab:tot}): while the SUB scheme tends to overestimate the magnitude and extension of total cloud amounts across the ITCZ, the MIC scheme shows a slight underestimation In addition, use of the MIC scheme improves the stratiform cloud cover between 30 and 45° S, yielding higher spatial correlation values compared to those obtained with SUB."

Line 325: The authors might add "GCM-oriented" CALIPSO estimates to be more specific.

Text modified: "The GCM-Oriented CALIPSO Cloud Product GOCCP data (Chepfer et al., 2010), 2° x 2°, are used for the model evaluation as they are designed for comparisons with output from the CALIPSO satellite simulator."

Line 326: Please use a more recent reference for CALIPSO: Winker et al., 2010, doi:10.1175/2010BAMS3009.1

Done.

Line 327: Which version of CALIPSO-GOCCP did you use and what about the resolution and the time period? Judging from the figure, it seems to be only one season and 1degx1deg grid. I would strongly encourage the author to at least pick the 2x2deg grid and averaged over all available seasons to smooth the noise. As mentioned before, the inter-annual variation is lower than the mod-to-obs bias anyway and should change

the pattern of the bias. The 1x1deg grid is also very noisy because of the poor overlap due to CALIPSO polar orbit.

We used the 2x2 deg grid (added in the text) and averaged over the same season simulated by the model, following Franklin (2013). In Fig. 1 we show that the average cloud amount for the available climatology (JJA 2006-2010) does not substantially differ from the chosen season (JJA 2007), suggesting that the inter-annual variation in global mean high, medium and low total cloud fractions is not large and it is reasonable to choose only one season:

Line 346: liquid droplets rather than cloud droplets.

Done.

Table 3: same as for Tab 2, SUB ! MIC ! OBS and maybe the authors should adding correlation numbers.

Done.

For MISR, same questions as for ISCCP and GOCCP, are these from CFMIP-obs? And what is the resolution?

The following text was modified: "An even more accurate analysis of cloud vertical distribution can be carried out with the use of the Multi-angle Imaging SpectroRadiometer MISR (Muller et al. 2002) data. MISR uses nine cameras providing images with approximately 275 m sampling in four narrow spectral bands, spanning much of the angle range over which cloud reflectivity varies. This leads to a more accurate retrieval of albedo than the use of a single camera. Naud et al. (2002), however, found that in the case of multi layered clouds MISR often "sees" through the thin upper level clouds and mostly refers to low level cloud layers. The MISR retrievals can be processed to produce joint histograms of Cloud Top Height (CTH) and Optical Depth (OD) used specifically for a comparison with the COSP output and available on the CFMIP observational dataset website. To compare with the MISR retrievals, we postprocessed the

RegCM4 data with the MISR simulator described in Marchand et al. (2010)."

Line 367-369: May these low clouds be the shallow cumulus cloud, implying that the RegCM4 model struggle to represent the transition from strato to shallow cumulus clouds, as many other models? Besides, it is in agreement with the few too bright problem, too few low clouds but optically too thick (e.g. Nam et al., 2012, doi:10.1029/2012GL053421)

We added the following text: "While an underestimation of low clouds is a common problem in climate models (e.g. Nam et al., 2012; Zhang et al., 2005) a reason for the overestimation of thick clouds found here may reside in the fact that even if in real systems only part of a 100 km grid area experiences a strong upward motion, the mean vertical velocity for the whole model grid box is upward, leading to an updraft for the entire gridbox. A reason for the overestimation of low optically thick clouds can be related to the coarse horizontal model resolution (100 km) which does not resolve surface-heterogeneity, topography and shallow mesoscale circulations. Future studies will evaluate the model performance at higher horizontal resolutions." Section 3.3

Did the authors use the CERES-EBAF data, specially designed for model evaluation? Please, clarify and define the resolution and time period.

We used the CERES ERBA-like Monthly Geographical Averages (ES-4) observations (Wielicki, 2011), with an horizontal resolution of 2.5° x 2.5° (added in the text) because we wanted to compare our results with those observed in the same season. CERES-EBAF are only available until 2005. While the explanation for the CRFlw is straightforward, the upper cloud issue does not explain all of the CRFsw bias, and the authors do not refer to the other reasons of the bias. As mentioned before, the low clouds have been shown to be mostly too reflective in many GCMs for quite a while now (e.g. Nam et al., 2012; Zhang et al., 2005). It seems to be also the case for RegCM4. The large bias in the CRFsw remains even in region where the upper and lower clouds are well reproduced by the model (e.g. around 45°S). Could

this be because of the thin low clouds missed by your model as shown by the MISR simulator analysis? Could you i) locate these thin low clouds on a map and ii) propose an explanation of why they are so optically thick? We modified the following text: "The excessive upper level cloud cover in the SUB run yields too much SW reflection and therefore the domain-average SW values are about 40 W m$^{-2}$ lower than observed. The MIC scheme, by reducing the upper level cloud cover, reduces the upward SW flux and therefore yields values closer to observations (domain average bias of 10 W m$^{-2}$). However a substantial bias still persists in areas where both high and low clouds are well represented (e.g. around 45° S). This bias can be attributed to the underestimation of thin low clouds as shown by the MISR simulator analysis (Figure \ref{misr}). Even if the overestimation of low cloud reflectivity is a common problem for many GCMs \citep[e.g.][]{nam2012too,zhang2005} a reason for our overestimation of low optically thick clouds can be related to the coarse horizontal resolution (100 km) which does not resolve surface-heterogeneity, topography and shallow mesoscale circulations. Future studies will evaluate the model performance at higher horizontal resolutions." When looking at the full CRF, i.e. the sum of CRF$_{SW}$ and CRF$_{LW}$ (Figure \ref{fig:cretot} and Table \ref{tab:cre}), we see that essentially the model biases tend to compensate, yielding values close to each other for the two schemes and not far from observations (although on a domain average the MIC is still closer to observations by a few W/m2). In some tropical monsoon regions the longwave gain in the SUB scheme appears to be larger than the shortwave loss, leading to an overall heating which is less pronounced in the MIC scheme.

Line 418: cirrus instead of stratocumulus.

Done
* * *
[Figure]

Fig. 1. Same as Fig 4. in the paper with modified panel g) h) and i) that now show observations for the available climatology (JJA 2006-2010).

---

## Author Comment (AC3) · 7 Jun 2016

First, we would like to thank all the reviewers for their careful reviews and constructive comments, which helped to improve the quality and clarity of the paper.

Anonymous Referee #3

The paper introduces a needed update to the moist physics in the RegCM4 community regional climate model, namely the inclusion of ice phase microphysics. Given the wide use of RegCM4 it is likely that this paper will be heavily referenced. The paper is well written and there are only a few minor changes needed to clarify and strengthen it.

1. There are many microphysical schemes in existence, some of which are more detailed than the scheme here and some less. It would be appropriate to discuss briefly the rationale for choosing this particular scheme for inclusion in RegCM4 compared to other options.

Essentially, we chose this particular microphysics scheme because of its robust fully implicit numerical framework that allows the use of longer timesteps and because it is based on the scheme used and widely tested in the ECMWF IFS forecasting system. This clarification is specified in the text in lines 105-109.

2. At line 68: Is there no rainwater evaporation in SUBEX?

Yes, SUBEX treats the rainwater evaporation, but being diagnostic it is considered to have an infinite fall speed and can not be advected.

3. Are any of the parameters in the new scheme known or suspected to be sensitive to grid spacing? Intuitively it would seem that some of the parameters (such as those in Equation 5) should approach limiting values for very small grid volumes and as such their most appropriate values could vary with grid spacing.

This comment is well taken. We have not yet carried out a full sensitivity analysis to model resolution, in particular for very high resolutions, which is in fact planned as the next step.

4. Line 195, "condensate" should be "condense."

Done.

5. The RHS of equation (12) simply works out to D, since alpha + (1-alpha) = 1. This does not seem correct. Are there missing subscripts or other corrections needed?

We thank the reviewer for the comment: there was a mistake in the equation, now corrected as follows:

$$(\partial q\_x)/\partial t = \alpha(T)D\_x$$

6. Line 205, regarding the four different autoconversion parameterizations: Are these user-selectable, or are different parameterizations invoked automatically by the scheme depending on the physical conditions?

The autoconversion parameterizations can be selected by the user. dded in the text: " The four parameterizations of autoconversion in the scheme, which can be selected by the user, employ different threshold functions: an "all-or-nothing" approach, described in ..."

7. Equation (14), the species for which ql and qcrit apply should be clarified. Typically the rate on the LHS applies to precipitation and the humidity on the right-hand side is cloud water, but this equation has ql on both the LHS and RHS implying a positive feedback (which seems unusual).

Done.

8. Line 244, The reference on IFS documentation does not appear in the list of references, or at least not under that title. Please give sufficient bibliographic information so that the reader can access this document.

We added in the text: " For a more detailed description of the parameterization of microphysical processes we refer the reader to the IFS Documentation, Cy40r1, Part IV: Physical Processes (online at https://software.ecmwf.int/wiki/display/IFS/Official+IFS+Documentation)."